# Rspo3-mediated metabolic liver zonation regulates systemic glucose metabolism and body mass in mice

Kenji Uno[1]*, Takuya Uchino[1], Takashi Suzuki[1], Yohei Sayama[1], Naoki Edo[2], Kiyoko Uno-Eder[2], Koji Morita[1], Toshio Ishikawa[1], Miho Koizumi[3], Hiroaki Honda[3], Hideki Katagiri[4], Kazuhisa Tsukamoto[1]

1 Department of Internal Medicine, Teikyo University School of Medicine, Tokyo, Japan, 2 Teikyo Academic Research Center, Tokyo, Japan, 3 Field of Human Disease Models, Tokyo Women's Medical University, Tokyo, Japan, 4 Department of Metabolism and Diabetes, Tohoku University Graduate School of Medicine, Sendai, Japan

* unok@med.teikyo-u.ac.jp, unok@h4.dion.ne.jp

## Abstract

The unique architecture of the liver consists of hepatic lobules, dividing the hepatic features of metabolism into 2 distinct zones, namely the pericentral and periportal zones, the spatial characteristics of which are broadly defined as metabolic zonation. R-spondin3 (Rspo3), a bioactive protein promoting the Wnt signaling pathway, regulates metabolic features especially around hepatic central veins. However, the functional impact of hepatic metabolic zonation, regulated by the Rspo3/Wnt signaling pathway, on whole-body metabolism homeostasis remains poorly understood. In this study, we analyze the local functions of Rspo3 in the liver and the remote actions of hepatic Rspo3 on other organs of the body by using murine models. Rspo3 expression analysis shows that Rspo3 expression patterns are spatiotemporally controlled in the murine liver such that it locates in the pericentral zones and converges after feeding, and the dynamics of these processes are disturbed in obesity. We find that viral-mediated induction of Rspo3 in hepatic tissue of obesity improves insulin resistance and prevents body weight gain by restoring attenuated organ insulin sensitivities, reducing adipose tissue enlargement and reversing overstimulated adaptive thermogenesis. Denervation of the hepatic vagus suppresses these remote effects, derived from hepatic Rspo3 induction, toward adipose tissues and skeletal muscle, suggesting that signals are transduced via the neuronal communication consisting of afferent vagal and efferent sympathetic nerves. Furthermore, the non-neuronal inter-organ communication up-regulating muscle lipid utilization is partially responsible for the ameliorations of both fatty liver development and reduced skeletal muscle quality in obesity. In contrast, hepatic Rspo3 suppression through Cre-LoxP-mediated recombination system exacerbates diabetes due to glucose intolerance and insulin resistance, promotes fatty liver development and decreases skeletal muscle quality, resulting in obesity. Taken together, our study results reveal that modulation of hepatic Rspo3 contributes to maintaining systemic glucose metabolism and body composition via a newly identified inter-organ communication mechanism.

**Data Availability Statement:** The authors confirm that all data underlying the findings are fully available without restriction. All relevant data are within the paper and its Supporting Information

files. The original data for the RNA-sequencing experiments have been deposited on the Gene Expression Omnibus (GEO accession GSE282890). Metabolomics data has been deposited to the Metabolomics Workbench (accession number ST003586).

**Funding:** The studies described were supported by the Ministry of Education, Culture, Sports, Science and Technology of Japan (KAKENHI numbers 18H02858, 22H03130, and 23K24389 to KU). KU has received research grants from Mochida Memorial Foundation for Medical and Pharmaceutical Research, Daiwa Securities Health Foundation, the TERUMO LIFE SCIENCE FOUNDATION, the Takeda Science Foundation, and the Koyanagi Foundation. The funders had no role in study design, data collection and analysis, decision to publish, or preparation of the manuscript.

**Competing interests:** The authors have declared that no competing interests exist.

**Abbreviations:** AA, amino acid; b2AR, b2-adrenergic receptor; b3AR, b3-adrenergic receptor; BAT, brown adipose tissue; BCAA, branched-chain amino acid; BCAT2, branched-chain AA aminotransferase 2; BCKDH, branched-chain ketoacid dehydrogenase; Cpt1a, carnitine palmitoyltransferase 1a; Cyp2f2, cytochrome P450 2f2; Cyp7a1, cholesterol 7alpha-hydroxylase; FASN, fatty acid synthetase; G1P, glucose-1-phosphate; G6P, glucose-6-phosphate; GK, glucokinase; GO, gene ontology; HE, hematoxylin eosin; HNF4a, Hepatocyte nuclear factor 4a; HRP, horseradish peroxidase; Hsd17, hydroxysteroid 17-beta dehydrogenase; HV, hepatic vagotomy; IF, immunofluorescence; IR, insulin receptor; ISH, in situ hybridization; NE, norepinephrine; Oat, ornithine aminotransferase; PAS, periodic acid–Schiff; PEPCK, phosphoenolpyruvate carboxykinase; PFA, paraformaldehyde; PPAR, peroxisome proliferator-activated receptor; Rdh9, retinol dehydrogenase9; RQ, respiratory quotient; rRPa, rostral raphe pallidus nucleus; Rspo3, R-spondin3; SCD1, stearoyl CoA desaturase 1; SO, sham operation; SREBP1c, sterol regulatory element-binding protein 1c; TG, triglyceride; WAT, white adipose tissue.

## Introduction

The unique architecture of the liver consists of hepatic lobules. These lobules have a distinct bloodstream construction, in which nutrient-rich venous blood from the intestines mixes with arterial blood around the portal triad, traveling through the hepatic sinusoids and collecting in central zones [1]. The lobular architecture divides hepatic features into 2 distinct metabolic zones, so-called "metabolic zonation" [2–4], which have spatially and temporally separate functions such as glucose (gluconeogenesis, glycolysis), lipid (β-oxidation, lipogenesis), and amino acid (AA) metabolism (glutamine synthesis). While mainly gluconeogenesis and β-oxidation are increased in the periportal zone, glycolysis, lipogenesis, and glutamine synthesis are increased in the pericentral zone [5–7]. Thus, metabolic zonation plays an important role in maintaining both intra-hepatic and systemic metabolism [8]. However, the precise regulatory mechanism underlying metabolic zonation in the liver and the pathophysiological significance of disturbances associated with nutrient overloading and their contributions to metabolic disorders, i.e., metabolic syndrome, have yet to be clarified.

Members of the R-spondin family (R-spondin1-4 (Rspo1-4)), proteins secreted by several organs, function as amplifiers of Wnt pathway activity by engaging LGR4-6 G protein-coupled receptors to increase the cell membrane density of Wnt receptors [6,9]. Wnts, also recognized as a family of secreted proteins, locally bind to their receptors, leading to nuclear accumulation of the transcription activator β-catenin and increases in downstream target gene expressions. This canonical pathway is regarded as an evolutionarily conserved cascade, playing important roles in stem cell biology and adult tissue homeostasis [10–12].

Furthermore, R-spondin3 (Rspo3) was demonstrated to be a key molecule determining metabolic zonation in the liver by regulating AA-related gene expressions such as glutamine synthase around central zones [13]. In addition to Rspo3 findings, Rspo1 was also identified as controlling metabolic zonation and hepatic size and growth via the RSPO-LGR4/5-ZNRF3/RNF43 module [14]. Overexpression or ablation of hepatic β-catenin, a downstream target of the Rspo family, in murine models on high-fat diet reportedly causes diet-induced obesity with progressive hepatic steatosis and insulin resistance or protects against diet-induced obesity, respectively [15]. In addition to these roles of the Rspo/Wnt/β-catenin signaling pathway in the liver, RSPO3 is associated with an increased body mass index-adjusted waist-to-hip ratio, impacting upper body fat distribution by stimulating abdominal adipose proliferation [16]. Ablation of LGR4, an Rspo family receptor, promoted energy expenditure and induced brown-like adipocytes in white adipose tissue (WAT) depots, i.e., white-to-brown transition [17]. These prior reports suggested that focusing on modulation of metabolic zonation in the liver via Rspo family proteins raises the possibility of elucidating inter-organ communication throughout the body.

Herein, we demonstrated that modulation of hepatic Rspo3 improves obesity-associated features, including diabetes, insulin resistance, and fatty liver. Most notably, hepatic Rspo3 reverses enlargement of WAT in obesity via a newly identified neuronal inter-organ linkage from the liver to adipose tissue. In contrast, hepatic Rspo3 suppression exacerbates obesity-associated features, including glucose intolerance, insulin resistance, and fatty liver change. Therefore, modulating and restoring metabolic zonation by targeting the hepatic Rspo3 signaling pathway may lead to novel therapeutic strategies for diabetes and metabolic syndrome.

## Results

### Metabolic functions of Rspo3 in the liver are disturbed in obesity

First, we examined whether serum Rspo3 levels were changed in either lean or genetically obese (ob/ob) murine models and found that Rspo3 levels did not differ significantly between

these 2 conditions in fasted states (Fig 1A). In lean models, but not in obesity, serum Rspo3 levels were up-regulated by feeding (Fig 1A). Second, we examined gene expression levels of Rspo3 in the liver (Fig 1B), adipose tissues including WAT and brown adipose tissue (BAT), and skeletal muscles of both lean and obese models (Fig 1C). We found that endogenous Rspo3 expressions in the liver and WAT, the major organs producing Rspo3, were decreased, while those in BAT and skeletal muscle were not, according to the degree of obesity. Feeding led to up-regulation of hepatic Rspo3 expressions in the livers of lean model mice, while no such phenomenon was observed in obese mice (Fig 1D). These results indicated that Rspo3 levels were dynamically altered according to both physiological conditions, including lean and obese states, and nutritional settings, such as fed and fasted states.

Next, we examined the localization of Rspo3 expression in hepatic lobules. To distinguish between the zones around central veins (pericentral) versus portal veins (periportal), we subjected liver sections to Azan staining, allowing the periportal zone to be detected (S1 Fig). Analysis of immunofluorescence (IF) of liver sections revealed much more Rspo3 expression in pericentral hepatocytes than in periportal hepatocytes (S1 Fig), suggesting the existence of zonation regulating hepatic Rspo3 expression and localization. Comparison of IF between Rspo3 and hepatocyte nuclear factor 4α (HNF4α, hepatocyte marker) confirmed that HNF4α-positive hepatocytes simultaneously possess Rspo3 IF reactivity (Fig 1E). Comparisons of IF between Rspo3 and cluster of differentiation 31 (CD31, endothelial cell marker) confirmed Rspo3-positive cells to be present at much lower densities in central vein endothelial cells than in hepatocytes (Fig 1E). Comparisons of IF between Rspo3 and cytokeratin 19 (CK19, sinusoidal endothelial marker) confirmed that there were no Rspo3-positive cells among neighboring sinusoidal endothelial cells (Fig 1E). Furthermore, Rspo3-positive hepatocytes in lean models were more responsive to stimulation or were increased in number, especially in the pericentral zone, in the fed than in the fasted state, and that the cellular dynamics of feeding on cumulative Rspo3 localization in the pericentral zone were disturbed in obese models (Figs 1F and S2). The localization of Rspo3-positive hepatocytes in obesity was rather widespread, extending from the pericentral zone to the periportal zone (Figs 1F and S2). In addition, the intensity of Rspo3 IF in hepatocytes in the pericentral zones, reflecting the extent of Rspo3 expression, indicated that feeding increased the intensity of Rspo3 IF, especially in lean models, while minimally impacting obese models (Fig 1G). Furthermore, to investigate the expression patterns of Rspo3 mRNA in the liver under these conditions such as lean versus obese states, or fasted versus fed states, we performed RNA-Scope, a newly developed technique of in situ hybridization (ISH), by using the specific probe against Rspo3. RNA-Scope revealed far more Rspo3-positive signals in the pericentral zones than in the periportal zones (Fig 1H), observations consistent with the results of IF (Fig 1F). In the pericentral zones, Rspo3-positive signals existed in endothelial cells as well as hepatocytes, reflecting the difference in Rspo3 expression patterns between mRNA and protein levels. On the other hand, the numbers of Rspo3-positive signals were increased in the pericentral zones of lean models after feeding, showing disturbance of the dynamics of feeding to change Rspo3 mRNA localization toward these zones, especially in obese models (Fig 1I).

Thus, the convergence of Rspo3-positive hepatocytes, at either the mRNA or the protein level, toward the pericentral zone in response to feeding (i.e., spatiotemporal characteristics of Rspo3) appears to play an important role in regulating intra-hepatic metabolism under normal conditions, a phenomenon which is dysregulated in pathophysiological states such as obesity.

## Rspo3 influences systemic glucose metabolism and body composition

To investigate whether hepatic Rspo3 functions in regulating metabolic physiology in mammals, we overexpressed Rspo3 in the livers of C57BL/6 mice fed normal chow diets

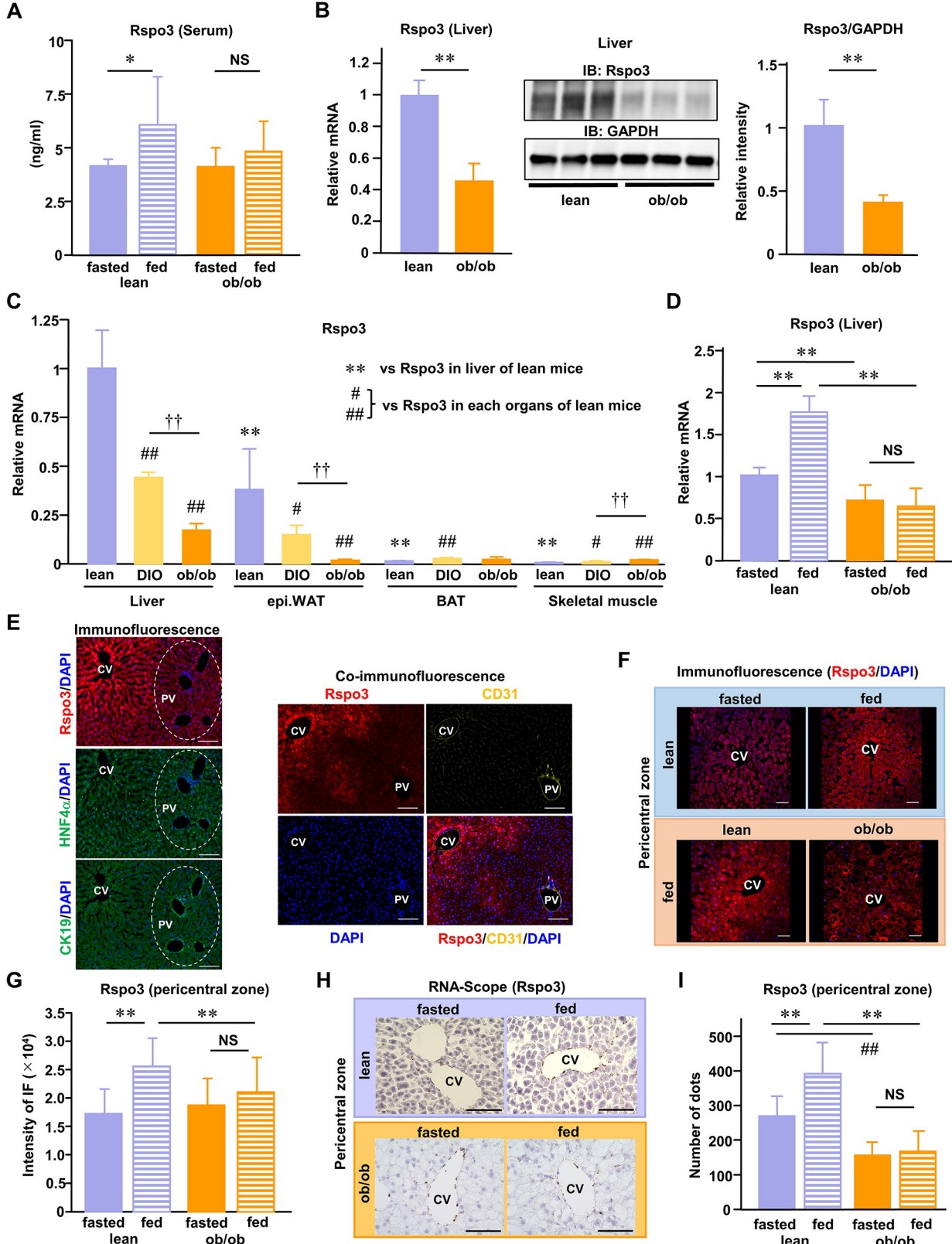

**Fig 1. Metabolic functions of Rspo3 in the liver are disturbed in obesity.** (A) Serum Rspo3 levels under either fasted or fed conditions were examined in lean and ob/ob mice ($n = 7$–14). (B) Hepatic Rspo3 mRNA expression ($n = 4$–5) and immunoblotting results of liver extracts with

anti-Rspo3 and anti-GAPDH antibodies were measured in lean and ob/ob mice. The relative intensities (the ratios of Rspo3/GAPDH protein volume) were calculated. **(C)** Rspo3 mRNA expressions in several organs were measured and compared among lean, DIO, and ob/ob mice ($n = 4$–8). **(D)** Hepatic Rspo3 mRNA expressions under either fasted or fed conditions were measured in lean and ob/ob mice ($n = 5$–13). **(E)** Representative immunofluorescence analysis images of the pericentral zone in the liver with anti-Rspo3, anti-HNF4α, anti-CD31, anti-CK19 antibodies, and DAPI staining. The scale bars indicate 100 μm. **(F)** Representative immunofluorescence analysis images of the pericentral zone in the liver with anti-Rspo3 antibody and DAPI staining. The scale bars indicate 50 μm. **(G)** The intensity of Rspo3 immunofluorescence was measured on 4 lines, each 20 μm from the hepatic central veins, obtained randomly (hepatic central veins; $n = 29$–30, 3 mice per group). **(H)** Representative RNA-Scope analysis images of the pericentral zone in the liver with specific Rspo3 probe. The scale bars indicate 100 μm. **(I)** The numbers of Rspo3-positive signals (dots) were measured randomly (hepatic central veins; $n = 13$–21, 3–4 mice per group). The data underlying the graphs shown in the figure can be found in **S1 Data**. Data are presented as means ± SD. $*P < 0.05$, $**P < 0.01$, $\#P < 0.05$, $\#\#P < 0.01$, $\dagger\dagger P < 0.01$ by the unpaired $t$ test. CD31, cluster of differentiation 31; CK19, Cytokeratin 19; DIO, diet-induced obese; HNF4a, Hepatocyte nuclear factor 4a; Rspo3, R-spondin3.

(Rspo3-mice) using adenoviral gene transduction. Systemic infusion of Rspo3 recombinant adenovirus resulted in selective transgene expression in the liver (1.46-fold of protein levels as compared to control mice) (Fig 2A), while there were no differences in Rspo3 expressions in other organs (S3A Fig). Mice given the lacZ adenovirus were used as controls (LacZ-mice). Hepatic Rspo3 induction significantly increased gene expressions of LGR5 and Axin2, downstream targets of the Rspo3/Wnt signaling pathway (S3B Fig). While hepatic Rspo3 induction increased gene expressions of ornithine aminotransferase (Oat), cholesterol 7alpha-hydroxylase (Cyp7a1), and retinol dehydrogenase9 (Rdh9) [13], which are expressed and function mainly in the pericentral zones (S3C Fig), it decreased hydroxysteroid 17-beta dehydrogenase (Hsd17) and cytochrome P450 2f2 (Cyp2f2) [13], which are expressed and function mainly in the periportal zones (S3D Fig). These results indicated hepatic Rspo3 induction to up-regulate the Rspo3/Wnt signaling pathway in the liver and thereby shifted metabolic gene expressions, a process by which hepatic features of metabolism are considered to move from the periportal to the pericentral zone. Under these conditions, hepatic Rspo3 expression significantly decreased fasting blood glucose and insulin levels (Fig 2B), producing both a decrease in the expression of a major hepatic gluconeogenic enzyme, phosphoenolpyruvate carboxykinase (PEPCK) and an increase in the expression of the hepatic glycolytic enzyme glucokinase (GK) (Fig 2C). While hepatic expressions of lipogenic genes such as sterol regulatory element-binding protein 1c (SREBP1c), fatty acid synthetase (FASN), and peroxisome proliferator-activated receptor (PPAR) γ, were increased, hepatic expressions of lipogenic genes such as stearoyl CoA desaturase 1 (SCD1) were decreased in Rspo3-mice (Fig 2D). Hepatic expressions of lipogenic and lipid utilizing genes, such as PPARδ, and lipid oxidative genes, such as carnitine palmitoyltransferase 1a (Cpt1a), but not PPARα, were also increased in Rspo3-mice (Fig 2D). Under these conditions, hepatic Rspo3 expression altered liver weight (Fig 2E) with no change in liver histology, as demonstrated by hematoxylin eosin (HE) staining (Fig 2F). In agreement with previous reports showing Rspo3 to have cell proliferative capacity [14,18], hepatic Rspo3 induction increased Ki67 mRNA levels 10- or 5-fold compared to those of control mice on day 7 or 23 after adenovirus administration, respectively (S3E Fig). These results suggested that hepatic Rspo3 induction dramatically altered the types of gene expressions in the liver (shown in Fig 2C and 2D), some of which were still present on day 23 after adenovirus administration (S3F Fig), giving rise to the feature of localization in the periportal zone toward a more pericentral zone. Glucose and insulin tolerance tests revealed that Rspo3-mice exhibited increased systemic glucose tolerance and insulin sensitivity, respectively (Fig 2G and 2H), the beneficial effects of which persist for 20 days after adenovirus administration (S3G and S3H Fig). To explore whether hepatic Rspo3 induction influences whole-body energy expenditure, we examined oxygen consumption (VO$_2$) and the respiratory quotient (RQ), finding no VO$_2$ difference while RQ was decreased in Rspo3-mice (Fig 2I). Furthermore, hepatic Rspo3 induction significantly decreased the weight of epididymal WAT (Fig 2J) due to increased β3-adrenergic

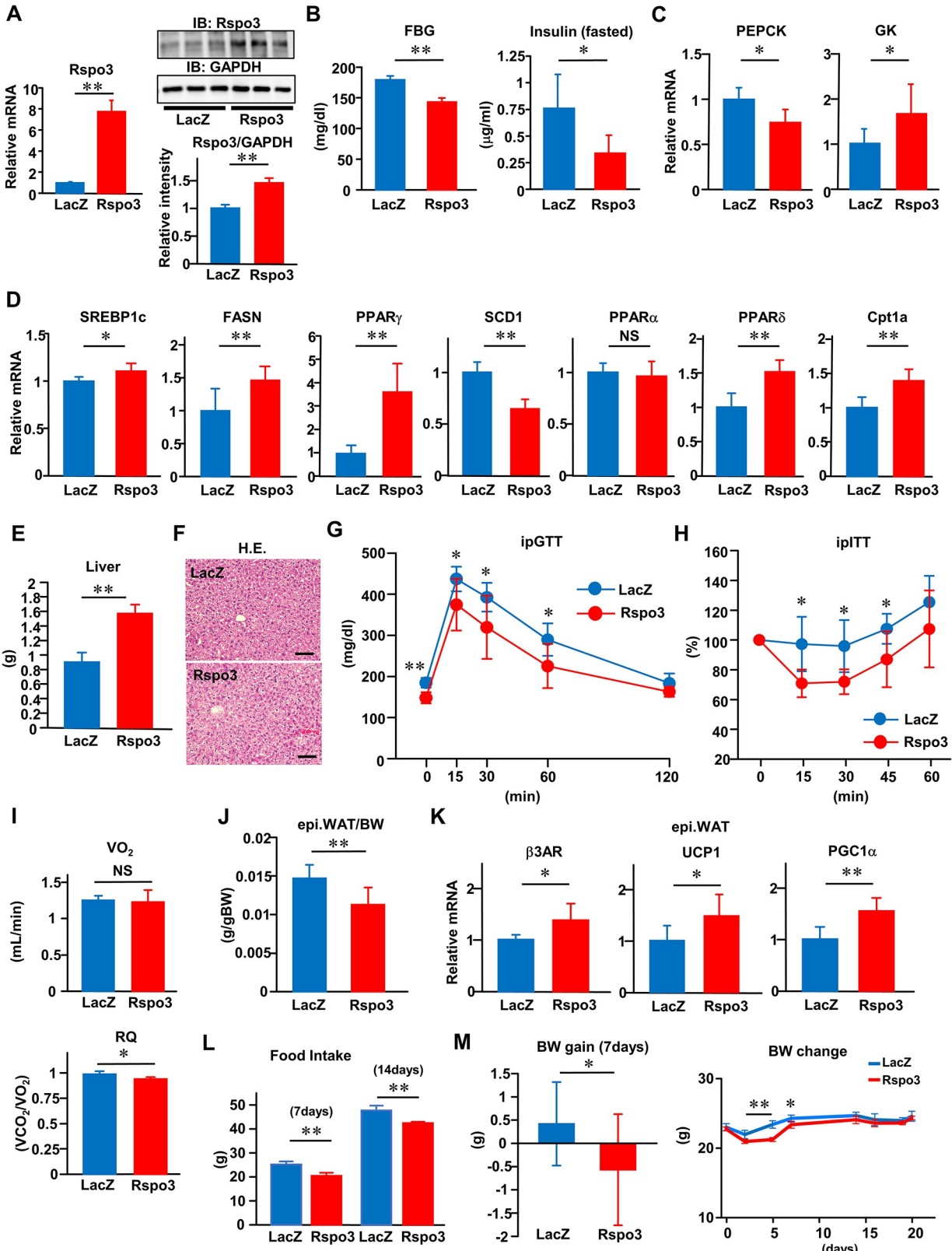

**Fig 2. Rspo3 influences systemic glucose metabolism and body composition.** (A–M) Rspo3 (red bars and circles) or LacZ (blue bars and circles) adenovirus was administered to standard diet-fed control mice. (A) Hepatic Rspo3 mRNA expression (*n* = 5) and immunoblotting of

liver extracts with anti-Rspo3 and anti-GAPDH antibodies and the relative intensities (the ratios of Rspo3/GAPDH protein volume), **(B)** fasting blood glucose ($n = 4–5$) and insulin levels ($n = 5–9$), **(C)** hepatic PEPCK and GK mRNA expressions ($n = 5–7$), **(D)** hepatic SREBP1c, FASN, PPARγ, SCD1, PPARα, PPARδ, and Cpt1a mRNA expressions ($n = 7$), **(E)** liver weight ($n = 6–7$), and **(F)** representative histological analysis images of the liver (HE staining) were examined on day 7 after adenovirus administration. The scale bars indicate 100 μm. **(G)** Glucose tolerance ($n = 5–11$) and **(H)** insulin tolerance ($n = 6$) tests were performed on day 7 after adenovirus administration. **(I)** VO$_2$ and RQ ($n = 5–6$) were measured on day 4 after adenovirus administration. **(J)** Epididymal WAT weight ($n = 7$), **(K)** β3AR, UCP1, and PGC1α mRNA expressions in epididymal WAT ($n = 5–7$); **(L)** food intake ($n = 5$); and **(M)** body weight gain ($n = 11–12$) were examined on day 7 after adenovirus administration. Body weight changes from day 0 to day 20 after adenovirus administration were measured ($n = 5$). The data underlying the graphs shown in the figure can be found in **S1 Data**. Data are presented as means ± SD. *$P < 0.05$, **$P < 0.01$ by the unpaired $t$ test. b3AR, b3-adrenergic receptor; Cpt1a, carnitine palmitoyltransferase 1a; FASN, fatty acid synthetase; FBG, fasting blood glucose; GK, glucokinase; HE, hematoxylin eosin; PPAR, peroxisome proliferator-activated receptor; PEPCK, phosphoenolpyruvate carboxykinase; RQ, respiratory quotient; Rspo3, R-spondin3; SCD1, stearoyl CoA desaturase 1; SREBP1c, sterol regulatory element-binding protein 1c; WAT, white adipose tissue.

receptor (β3AR), UCP1 and PGC1α expressions in epididymal WAT (Fig 2K) and/or decreased food intake (Fig 2L), contributing to the prevention of body weight gain for 7 days after adenoviral treatment (Fig 2M). Collectively, these results suggest hepatic Rspo3 to play an important role in the regulation of systemic glucose metabolism and body composition.

## Rspo3 improves obesity-induced metabolic disorders locally in the liver

Next, to investigate whether hepatic Rspo3 induction has any effects on obesity-related metabolic dysregulation, we administered Rspo3 adenovirus to obese (ob/ob) mice (obese Rspo3-mice). Systemic infusion of Rspo3 adenovirus resulted in selective transgene expression in the liver (1.85-fold of protein levels as compared to obese control mice), to the same or a degree slightly lower than that of endogenous Repo3 expression in lean control mice (0.74-fold) (Fig 3A). There were no differences in Rspo3 expressions in other organs (S4A Fig). RNA-Scope revealed that Rspo3-positive signals are also increased in obese Rspo3-mice (3.6-fold) (S4B Fig) to a lesser degree than indicated by the quantitative RT-PCR results (Fig 3A). However, hepatic Rspo3 induction did not alter circulating Rspo3 levels on day 7, showing only a temporary increase on day3, after adenovirus administration as compared to the level in control mice (S4C Fig). As in lean Rspo3-mice, hepatic Rspo3 induction significantly increased gene expressions of LGR5 and Axin2, downstream targets of the Rspo3/Wnt signaling pathway (S4D Fig), as well as gene expressions of Oat and Rdh9, markers of genes in the pericentral zone (S4E Fig), while it decreased Cyp2f2, a marker of genes in the periportal zone (S4F Fig). According to the RNA-sequence results, other markers of genes in the pericentral zone, such as Cyp2a4 (1.76-fold, $P < 0.01$), Cyp1a2 (2.00-fold, $P < 0.01$), Rnase4 (3.87-fold, $P < 0.01$), Cyp7a1 (1.19-fold, $P < 0.05$), Gstm3 (4.07-fold, $P < 0.01$), and Avpr1 (1.59-fold, $P < 0.05$), were increased, while other markers of genes in the periportal zone, such as Arg1 (0.75-fold, $P < 0.01$), Gls2 (0.20-fold, $P < 0.01$), Aldh1b1 (0.81-fold, $P < 0.01$), and Sds (0.28-fold, $P < 0.01$), were decreased in obese Rspo3 mice (S1 Table). All of these results suggested the reprograming of hepatocytes from periportal to pericentral features.

Meanwhile, comparing RNA-sequence results of lean mice with those of obese mice revealed that obesity actually alters several metabolic processes including glucose and lipid metabolism, as demonstrated by gene ontology (GO) studies and KEGG pathway analyses (S5 and S6 Figs). The gene expressions of LGR5, Axin2, Oat, Cyp7a1, and Rdh9 as well as those of other marker genes in the pericentral zone, such as Cyp1a2 (0.42-fold, $P < 0.01$), Rnase4 (0.30-fold, $P < 0.01$), Rdh9 (0.68-fold, $P < 0.01$), and Avpr1 (0.12-fold, $P < 0.01$), were decreased, while the marker gene in the periportal zone, Aldh1b1 (1.65-fold, $P < 0.01$), was increased in obese mice as compared to lean mice (S7A–S7C Fig and S2 Table). Considering the data obtained from models of hepatic Rspo3 induction in obese mice (S4D–S4F Fig and S1 Table), these results show obesity to influence the spatially regulated expression patterns of

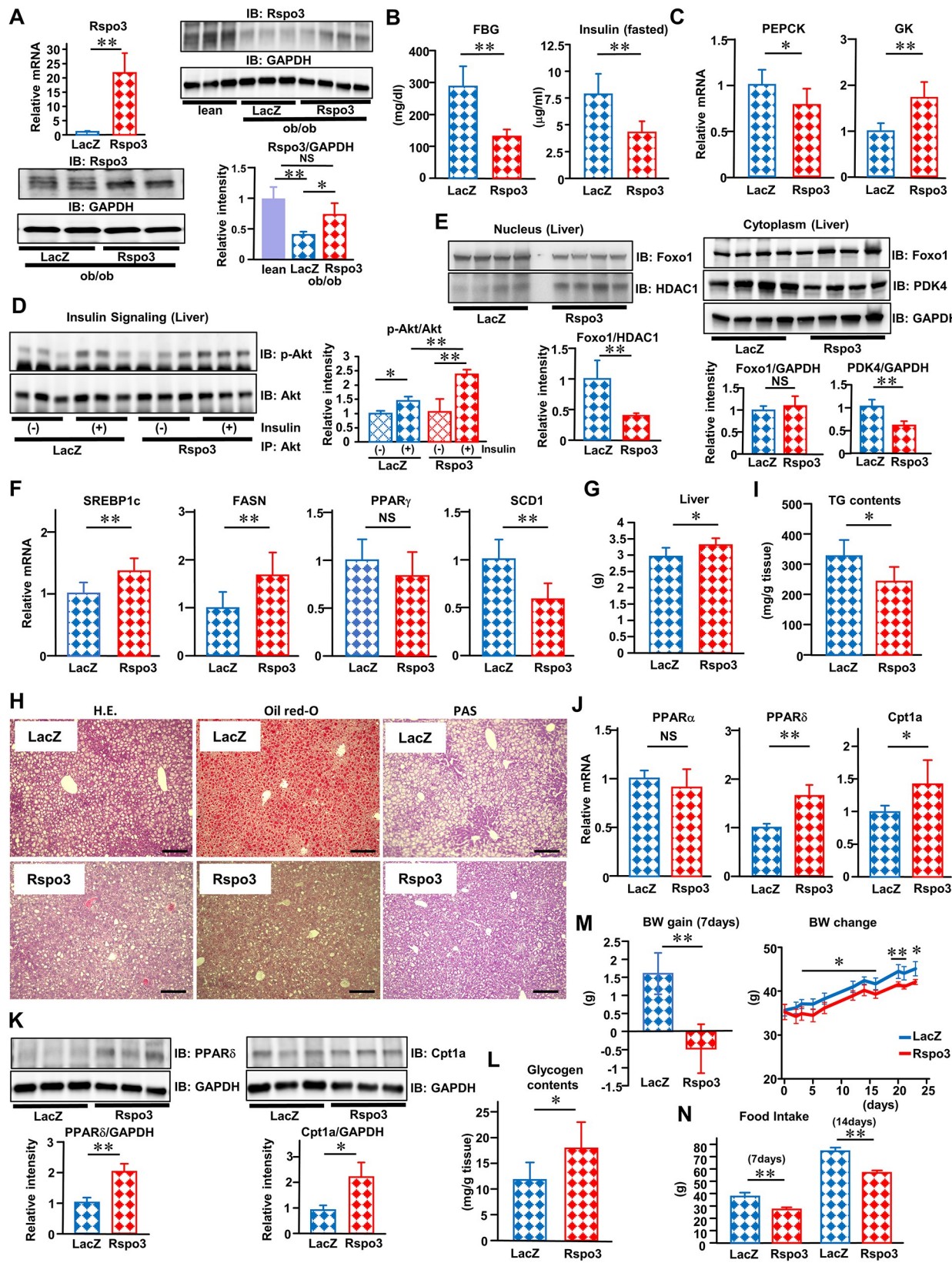

**Fig 3. Rspo3 improves obesity-induced metabolic disorders locally in the liver. (A–N)** Rspo3 (red bars) or LacZ (blue bars) adenovirus was administered to standard diet-fed ob/ob mice. **(A)** Hepatic Rspo3 mRNA expression ($n = 6$) and immunoblotting of liver extracts with anti-Rspo3 and anti-GAPDH antibodies and the relative intensities (the ratios of Rspo3/GAPDH protein volume), **(B)** fasting blood glucose and insulin levels ($n = 5$–7), and **(C)** hepatic PEPCK and GK mRNA expressions were examined on day 7 after adenovirus administration ($n = 6$–7). **(D)** Insulin-stimulated serine473 phosphorylation of Akt protein in the liver is presented. Mice fasted 16 h were injected intravenously with insulin or vehicle alone. The liver was removed 300 s later and lysates were immunoprecipitated with the indicated antibodies, followed by SDS-PAGE and immunoblotting with anti-phospho-Akt (Ser473) antibody, or just the antibody against Akt. The relative intensities (the ratios of p-Akt/Akt protein volume) were examined. **(E)** Immunoblotting of nuclear extracts with anti-Foxo1 and anti-HDAC1 antibodies and the ratios of Foxo1/HDAC1 protein volume; and cytoplasm extracts with anti-Foxo1, anti-PDK4, and anti-GAPDH antibodies; and the relative intensities (the ratios of Foxo1/GAPDH or PDK4/GAPDH protein volume); **(F)** hepatic SREBP1c, FASN, PPARγ, and SCD1 mRNA expressions ($n = 7$); and **(G)** liver weight were examined on day 7 after adenovirus administration ($n = 5$–7). **(H)** Representative histological analysis images of the liver, HE staining, Oil red-O staining, and PAS staining on day 5 after adenovirus administration. The scale bars indicate 200 μm. **(I)** Hepatic TG contents ($n = 6$–7); **(J)** hepatic PPARα, PPARδ, and Cpt1a mRNA expressions ($n = 7$); **(K)** immunoblotting of liver extracts with anti-PPARδ, anti-Cpt1a, and anti-GAPDH antibodies and the relative intensities (the ratios of PPARδ/GAPDH or Cpt1a/GAPDH protein volume); **(L)** hepatic glycogen content ($n = 7$); **(M)** body weight gain ($n = 5$–7) and body weight changes from day 0 to day 23 after adenovirus administration ($n = 4$–5); and **(N)** food intake ($n = 4$–5) were determined on day 7 after adenovirus administration. The data underlying the graphs shown in the figure can be found in S1 Data. Data are presented as means ± SD. *$P < 0.05$, **$P < 0.01$ by the unpaired $t$ test. Cpt1a, carnitine palmitoyltransferase 1a; FASN, fatty acid synthetase; FBG, fasting blood glucose; GK, glucokinase; HE, hematoxylin eosin; PAS, periodic acid chiff; PEPCK, phosphoenolpyruvate carboxykinase; PPAR, peroxisome proliferator-activated receptor; Rspo3, R-spondin3; SCD1, stearoyl CoA desaturase 1; SREBP1c, sterol regulatory element-binding protein 1c; TG, triglyceride.

marker genes in the pericentral and periportal zones of the liver. This pathophysiological phenomenon was suggested to be partially reversed by hepatic Rspo3 induction in these obese mice.

To investigate whether hepatic Rspo3 induction alters endogenous or exogenous Rspo3 localization in the liver on day 23 after adenovirus administration, we compared the staining patterns between Rspo3 IF, as a marker of endogenous plus exogenous Rspo3, and adenovirus-derived ZsGreen1 fluorescence, as a marker of exogenous Rspo3. Surprisingly, the results indicated much more adenovirus-derived Rspo3 to be present locally in the pericentral zones, leading to the expansion of Rspo3-positive hepatocytes, including endogenous or exogenous Rspo3, toward the periportal zones, as compared to those of control mice (S4G Fig). On the other hand, hepatic Rspo3 induction increased Ki67 IF intensity and Ki67 mRNA levels at this time point as compared to control mice (S4H and S4I Fig), observations compatible with those of previous reports describing Rspo3 as having cell proliferative capacity [14,18].

Metabolome analysis revealed hepatic Rspo3 induction as dramatically changing metabolic profiles, indicating that Rspo3 suppresses hepatic gluconeogenesis and increases glycolysis, glycogenesis, and synthesis of AAs (S8 and S9 Figs). RNA-sequence analysis also revealed that hepatic Rspo3 induction dramatically influenced several metabolic processes including glucose and lipid metabolism, as demonstrated by GO studies (S10 Fig). Under these conditions, hepatic Rspo3 induction significantly decreased fasting blood glucose and insulin levels (Fig 3B), while decreasing PEPCK expression and increasing GK expression (Fig 3C), indicating down-regulation of hepatic gluconeogenesis and up-regulation of glycolysis, respectively, results compatible with those of metabolome and RNA-sequence (PEPCK 0.66-fold, $P < 0.01$, and GK 1.54-fold, $P < 0.01$) analyses (S9 and S11 Figs). The examination of insulin signaling demonstrated hepatic Rspo3 induction to up-regulate insulin-stimulated serine473 phosphorylation of Akt in the livers of ob/ob mice (Fig 3D), which was also observed in another genetically obese (KK-Ay) mouse model (S12A Fig). Hepatic Rspo3 induction decreased Foxo1 expression in the nucleus (0.39-fold, $P < 0.01$) with no change of its expression in the cytoplasm of hepatocytes (Fig 3E). This restoration of hepatic insulin signaling resulted in increased Foxo1 exclusion out of the nucleus and decreased PDK4 expression in the cytoplasm of hepatocytes (0.59-fold, $P < 0.01$) (Fig 3E), leading to decreased hepatic gluconeogenesis [19] and increased hepatic glucose utilization in the TCA cycle [20], respectively. In support of the current research findings, previous studies demonstrated that insulin inhibits Foxo1 and

allows β-catenin to bind the Wnt signaling pathway effector TCF7L2, leading to down-regulation of PEPCK expression and thereby decreasing hepatic gluconeogenesis in the postprandial state [21].

Although hepatic Rspo3 induction up-regulated the expressions of SREBP1c and FASN (Fig 3F) as well as those of FASN (1.51-fold, $P < 0.01$) and other lipogenic genes examined by RNA-sequence analysis (S11 Fig), without changing PPARγ expression (Fig 3F), and slightly increased liver weight (Fig 3G), histological analysis of HE and Oil red-O stained specimens revealed the size and number of lipid droplets to be decreased in Rspo3-mice (Fig 3H), observations compatible with the decreased hepatic triglyceride (TG) contents in obese Rspo3 mice (Fig 3I). Interestingly, RNA-sequence analysis revealed that hepatic expressions of PPARδ (1.31-fold, $P < 0.05$), which regulates the de novo lipogenic pathway in the liver and fatty acid utilization in skeletal muscle [22], and Cpt1a (1.21-fold, $P < 0.05$), which regulates fatty acid oxidation and mitochondrial biogenesis in the liver [23], were elevated in obese Rspo3-mice (S11 Fig). Previous studies demonstrated that Wnt/β-catenin signaling serves as an upstream regulator for PPARδ expression [24] and drives fatty acid oxidation in dendritic cells by up-regulating Cpt1a expression [25]. This observation was followed by confirmation of elevations of mRNA expressions of PPARδ and Cpt1a, but not of PPARα (Fig 3J), as well as protein expressions of PPARδ (1.99-fold, $P < 0.01$) and Cpt1a (2.39-fold, $P < 0.05$), in obese Rspo3-mice (Fig 3K), partially accounting for the amelioration of obesity-related hepatic steatosis. In addition, according to RNA-sequence analysis, hepatic expression of SCD1, a lipogenic enzyme catalyzing the synthesis of monounsaturated fatty acids such as oleate and palmitoleate, was decreased (0.57-fold, $P < 0.01$) (S11 Fig), and this was followed by confirmation of decreased SCD1 expression in obese Rspo3-mice (Fig 3F). As previously reported, SCD1 deficiency increased fatty acid oxidation through AMPK activation and protected against fatty liver development induced by high-fat diet feeding [26], suggesting that decreased SCD1 expressions might have contributed to the amelioration of fatty liver findings in obese Rspo3-mice. Thus, hepatic Rspo3 induction, also in obese mice, was demonstrated to dramatically alter the types of gene expressions in the liver (shown in Fig 3C, 3F, and 3J), some of which were still present on day 23 after adenovirus administration (S12B Fig).

Furthermore, periodic acid–Schiff (PAS) staining results suggested that hepatic Rspo3 expression increased glycogen storage in the liver (Fig 3H). Thus, we calculated hepatic glycogen contents, which indicated that hepatic glycogen storage was increased in obese Rspo3-mice (Fig 3L), a phenomenon consistent with the profile obtained from metabolome analysis. In detail, metabolome analysis revealed the glucose-6-phosphate (G6P) level to be increased (1.8-fold, $P < 0.01$) and the conversion of G6P to glucose-1-phosphate (G1P), the first step of the glycogen synthesis pathway in the liver, was also up-regulated (1.2-fold, $P < 0.05$) in obese Rspo3-mice, followed by an increase in UDP-glucose synthesis (1.7-fold, $P < 0.01$) from G1P, the next step of this pathway (S8 and S9 Figs). Based on these results, we can reasonably speculate that hepatic Rspo3 induction locally diminishes obesity-related fatty liver progression by balancing the degrees of lipogenesis, lipid oxidation, and glucose utilization (gluconeogenesis, glycolysis, and glycogenesis) in the liver (S12C Fig).

Interestingly, hepatic Rspo3 induction suppressed obesity-induced body weight gain for 7 and 23 days with decreased cumulative food intakes for 7 and 14 days after adenoviral treatment (Fig 3M and 3N). Therefore, Rspo3, locally in the liver, functions as a key molecule suppressing the progression of obesity-related fatty liver, and possesses an inherent ability to influence the systemic management of body composition, leading to maintenance of glucose metabolism at the whole-body level.

## Rspo3 ameliorates the systemic glucose metabolism and body composition abnormalities in obese mice

As for glucose metabolism, hepatic Rspo3 induction decreased fasting blood glucose and serum insulin levels (Fig 3B), observations consistent with both decreased expression of PEPCK and increased expression of GK in the liver (Fig 3C). Glucose tolerance and insulin tolerance tests indicated that hepatic Rspo3 induction improved obesity-induced glucose intolerance and insulin resistance, respectively, in ob/ob mice (Fig 4A and 4B), observations also made in DIO and KK-Ay mice (S13A–S13C Fig). These improvements of glucose tolerance by hepatic Rspo3 induction were also observed in obese mice on day 20 after adenovirus administration (S13D and S13E Fig), with the beneficial effects persisting for 20 days. Specifically, the examination of insulin signaling demonstrated that hepatic Rspo3 induction up-regulated insulin-stimulated tyrosine phosphorylation of the insulin receptor (IR) or serine473 phosphorylation of Akt in remote organs, such as skeletal muscle and epididymal WAT (Fig 4C), as well as in the livers (Fig 3D), of ob/ob mice. Improvements of insulin signaling in remote organs were also more evident in KK-Ay mice (S13F Fig). These results indicated that hepatic Rspo3 induction improved obesity-induced insulin resistance throughout the body, leading to amelioration of diabetes.

Next, we examined whether hepatic Rspo3 expression has any effects on obesity-induced derangements of body composition including adipose tissue and skeletal muscle. Hepatic Rspo3 induction suppressed body weight gain (Fig 3M) as well as reducing epididymal WAT weight (Fig 4D) with increased β3AR, UCP1, and PGC1α expressions (Fig 4E), suggesting increased sympathetic nerve activity to WAT thereby raising WAT-derived thermogenesis, i.e., promotion of the transition from white to beige adipocytes [27,28]. This observation was also supported by the results of norepinephrine (NE) turnover experiments, which revealed that hepatic Rspo3 induction increased the rate of the NE decrease in epididymal WAT after treatment with α-MT, a catecholamine synthesis inhibitor, suggesting the up-regulation of sympathetic stimulation to WAT with resultant increases of both NE production and turnover (Fig 4F). In contrast, β3AR, UCP1, and PGC1α expressions in BAT were all decreased in obese Rspo3-mice (Fig 4G). As indicated by a previous report showing branched-chain AA (BCAA) metabolism to be necessary for BAT-derived thermogenesis [29], BAT gene expressions of BCAA-catabolizing enzymes such as branched-chain AA aminotransferase 2 (BCAT2) and branched-chain ketoacid dehydrogenase (BCKDH) were decreased in obese Rspo3-mice (Fig 4G). Next, to examine the neural activity in the rostral raphe pallidus nucleus (rRPa), containing the putative sympathetic premotor neurons responsible for thermogenesis in BAT, we performed c-fos IF analysis of the rRPa sections of the brains of these obese mice. The IF results indicated that hepatic Rspo3 induction significantly decreases c-fos IF intensity in these areas, suggesting the suppression of sympathetic activity to BAT (S14 Fig).

Furthermore, to explore whether hepatic Rspo3 induction in obese mice exerts an influence on whole-body energy expenditure, we examined $VO_2$ and RQ, demonstrating decreased $VO_2$ and decreased RQ in obese Rspo3 mice (Fig 4H). In addition, we examined whether there are any influences on $VO_2$ after NE treatment in obese Rspo3 mice, and obtained evidence indicating that hepatic Rspo3 induction decreases the degree of NE-induced $VO_2$ elevation (Fig 4I). Obesity induced endogenous UCP1 expression in BAT and WAT in murine models (S15A and S15B Fig), resulting in increased resting $VO_2$ as compared to that of lean mice (S15C Fig). While adipocyte size in epididymal WAT was reduced, adipocytes in BAT were enlarged in obese Rspo3-mice, and this was most clearly observed in KK-Ay mice (Fig 4J). These results suggested that Rspo3 induction in the setting of obesity restores the hyperactivated metabolic rates of BAT-derived adaptive thermogenesis against the development of

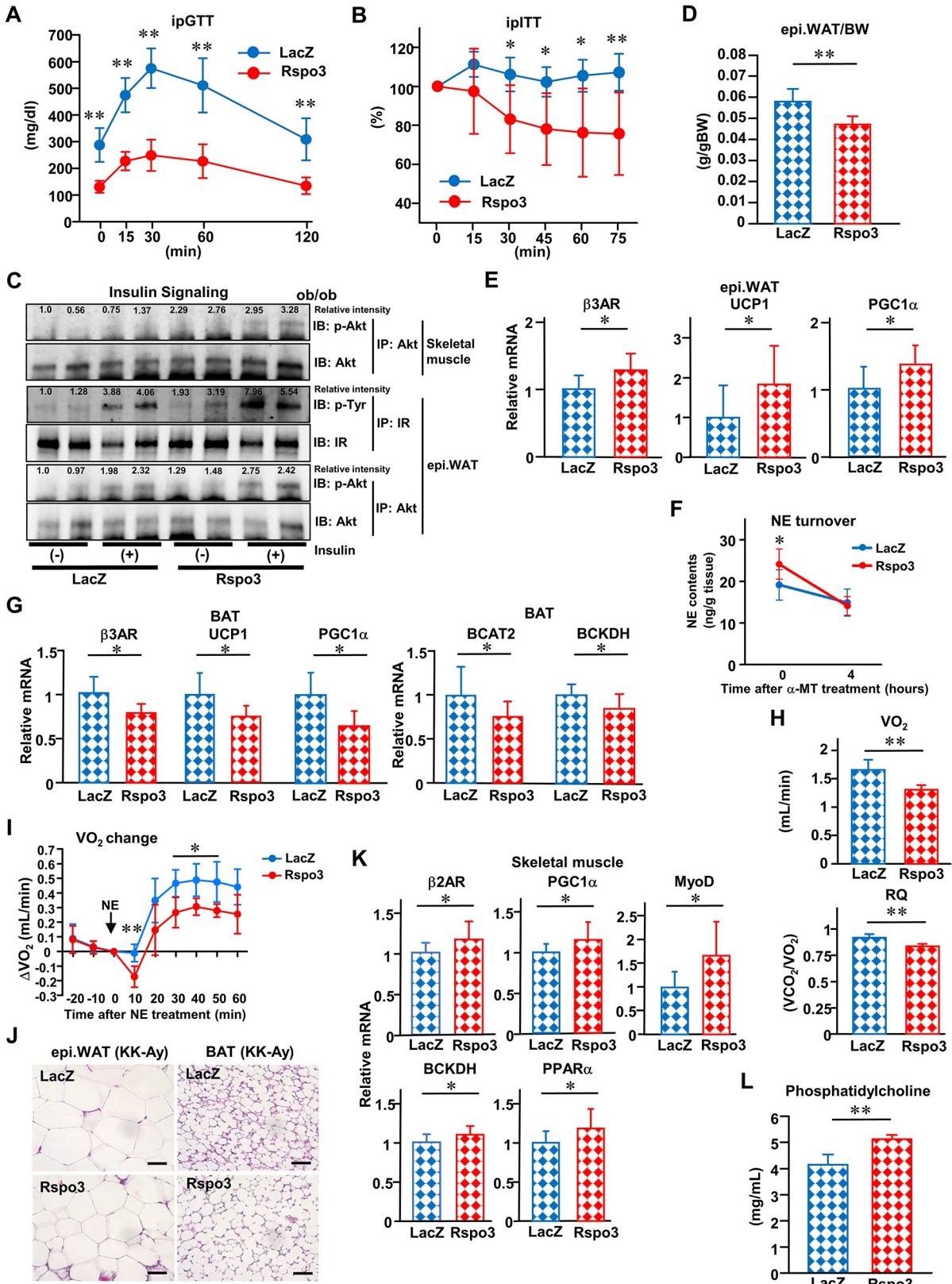

**Fig 4. Rspo3 ameliorates the systemic glucose metabolism and body composition abnormalities in obese mice. (A–L)** Rspo3 (red bars and circles) or LacZ (blue bars and circles) adenovirus was administered to standard diet-fed ob/ob or KK-Ay mice. **(A)** Glucose tolerance

and (B) insulin tolerance tests were performed on day 7 after adenovirus administration ($n$ = 5–7). (C) Insulin-stimulated serine473 phosphorylation of Akt protein in skeletal muscle (hindlimb muscles) and WAT were immunoblotted with anti-phospho-Akt (Ser473) antibody, or just the antibody against Akt. Insulin-stimulated tyrosine phosphorylation of IR protein in WAT was immunoblotted with anti-phospho-Tyrosine antibody, or just the antibody against IR. The relative intensities (the ratios of p-Akt/Akt or p-Try/IR protein volume) were indicated in the p-Akt or p-Tyr panels, respectively. (D) Epididymal WAT weight ($n$ = 5–7) and (E) β3AR, UCP1, and PGC1α expressions in epididymal WAT (β3AR; $n$ = 6–7, UCP1; $n$ = 12–13, PGC1α; $n$ = 7) were examined on day 7 after adenovirus administration. (F) NE turnover in WAT of KK-Ay mice ($n$ = 6–8) was performed on day 4 after adenovirus administration. (G) β3AR, UCP1, PGC1α, BCAT2, and BCKDH mRNA expressions in BAT (β3AR; $n$ = 7, UCP1, PGC1α; $n$ = 6–7, BCAT2, BCKDH; $n$ = 12–14) were examined on day 7 after adenovirus administration. (H) VO$_2$ and RQ ($n$ = 4–5) were measured on day 4 after adenovirus administration. (I) NE-induced VO$_2$ changes of KK-Ay mice ($n$ = 4–6) were measured on day 3 after adenovirus administration. (J) Representative histological analysis images of epididymal WAT and BAT (HE staining); (K) β2AR, PGC1α, MyoD, BCKDH, and PPARα mRNA expressions in skeletal muscle (β2AR, PGC1α, BCKDH, PPARα; $n$ = 12–13, MyoD; $n$ = 7); and (L) circulating phosphatidylcholine levels ($n$ = 7) were examined on day 7 after adenovirus administration. The scale bars indicate 50 μm. The data underlying the graphs shown in the figure can be found in **S1 Data**. Data are presented as means ± SD. *$P$ < 0.05, **$P$ < 0.01 by the unpaired $t$ test. a-MT; a-methyl-D,L-p-tyrosine, methyl ester, hydrochloride; b2AR, b2-adrenergic receptor; b3AR, b3-adrenergic receptor; BAT, brown adipose tissue; BCAT2, branched-chain AA aminotransferase 2; BCKDH, branched-chain ketoacid dehydrogenase; HE, hematoxylin eosin; IR, insulin receptor; NE, norepinephrine; PPAR, peroxisome proliferator-activated receptor; RQ, respiratory quotient; Rspo3, R-spondin3; WAT, white adipose tissue.

obesity, while it might be possible to activate WAT-derived thermogenesis. According to these effects, hepatic Rspo3 induction in obese states alters the balance of energy source use shifting toward utilization of lipids to a greater degree than carbohydrates, leading to maintenance of whole-body metabolism.

Hepatic Rspo3 induction increased β2-adrenergic receptor (β2AR) and resultant PGC1α expressions in skeletal muscle (Fig 4K) [30], thereby suggesting the restoration of adrenergic signaling, and skeletal muscle quality [31], which had been impaired in the setting of obesity. It also increased MyoD expression in skeletal muscle (Fig 4K), thereby suggesting the restoration of skeletal muscle remodeling and myogenesis [32], impaired in the setting of obesity. As described in a previous report indicating skeletal muscle PGC1α to regulate BCAA metabolism [33], skeletal muscle expression of BCAA-catabolizing enzymes such as BCKDH was increased in obese Rspo3-mice (Fig 4K). These results suggested that hepatic Rspo3 induction regulates the metabolism of BCAAs in remote organs such as skeletal muscle, thereby increasing the effective utilization of BCAAs, which results in maintenance of the physiological functions of skeletal muscle [34]. In addition, hepatic Rspo3 induction also increased PPARα expression in skeletal muscle (Fig 4K), leading to up-regulation of muscle lipid utilization and oxidation by cooperating with PGC1α [35]. A lipid-mediated inter-organ communication system linking the liver and skeletal muscle was previously identified, in which circulating fatty acids such as phosphatidylcholine (18:0/18:1), synthesized via the hepatic PPARδ-dependent de novo lipogenic pathway in the postprandial state, are utilized in skeletal muscle through its PPARα [22]. According to the results obtained by metabolome analysis, hepatic Rspo3 induction increased the levels of choline (1.8-fold, $P$ < 0.01) and phosphorylcholine (2.3-fold, $P$ < 0.01) or ethanolamine phosphate (2.8-fold, $P$ < 0.01) (S8 and S9 Figs), which are substrates for phosphatidylcholine synthesis through the Kennedy (CDP-choline) pathway or the methylation (phosphatidylethanolamine N-methyltransferase (PEMT)) pathway, respectively [36]. These results suggested that hepatic Rspo3 induction increases phosphatidylcholine synthesis in the liver, actually leading to the up-regulation of circulating phosphatidylcholine levels in obese Rspo3 mice (1.24-fold as compared to control mice) (Fig 4L). This raises the possibility that the inter-organ communication system operating under the hepatic PPARδ-phosphatidylcholine-muscle PPARα axis is involved, to a limited extent, in the ameliorations of both fatty liver and reduced skeletal muscle quality in obese Rspo3-mice. Therefore, these results indicated that hepatic Rspo3 induction improved the obesity-associated body composition imbalance between adipose tissue and skeletal muscle, leading to amelioration of the systemic metabolic disturbance.

## A neuronal inter-organ communication system is involved in Rspo3-induced remote effects

Next, we focused on the mechanism underlying this inter-organ communication system, in which hepatic Rspo3 induction in obesity improves glucose metabolism and alters body composition by diminishing obesity-related enlargement of WAT and by restoring sustained stimulation of adaptive thermogenesis (UCP1 elevation in BAT) in obesity. To address whether afferent neuronal signals from the liver are involved in these effects, we performed selective hepatic vagotomy (HV) [37] on an obese murine model (KK-Ay mice). There was no difference of body weight gain after sham or HV operations between sham operation (SO)-LacZ and HV-LacZ mice (S16A Fig), indicating that HV alone does not influence body composition, while HV nullified the body weight reduction obtained in response to hepatic Rspo3 induction. As for food intake, hepatic Rspo3 induction decreases the intakes of obese mice following either sham or HV operations (S16B Fig), indicating that HV alone does not influence the Rspo3-derived inhibitory effect on feeding. In obese Rspo3-mice, hepatic expression of Rspo3 and Rspo3-induced liver weight gains were unaffected by HV (Figs 5A and S16C). Hepatic expressions of genes associated with glucose metabolism such as PEPCK and GK (Fig 5B) as well as lipid metabolic enzymes such as FASN and PPARδ (Fig 5C) were essentially unaffected by HV. Rspo3-induced changes in the expressions of genes, such as LGR5, Axin2, Hsd17, and Cyp2f2, were also unaffected by HV (S16D and S16E Fig). In addition, Rspo3-induced alterations of Rspo3 IF pattern in the liver were unaffected by HV (S16F Fig). Under these conditions, HV almost completely blocked Rspo3-induced β3AR, UCP1 and PGC1α expression increases (Fig 5D) as well as Rspo3-induced WAT weight decrease and the reduction of white adipocyte size observed in SO-Rspo3 mice (Fig 5E and 5F). In addition, HV almost completely blocked Rspo3-induced β3AR, UCP1 and PGC1α expression decreases in BAT and the enlargement of brown adipocyte size observed in SO-Rspo3 mice (Fig 5G and 5H). As for BCAA metabolism, HV also almost completely blocked Rspo3-induced BCAT2 and BCKDH expression decreases in BAT (S16G Fig). Focusing on the NE contents in adipose tissues, HV almost completely blocked Rspo3-induced increases in the NE contents of WAT as well as the Rspo3-induced decrease in the NE contents of BAT (Fig 5I). Furthermore, HV almost completely blocked Rspo3-induced PGC1α and PPARα expression increases in skeletal muscle (Fig 5J). These results indicate a neuronal inter-organ communication system consisting of the hepatic vagus and sympathetic nerves to be involved in remote effects on adipose tissues and skeletal muscle, i.e., alterations of body composition, observed in Rspo3 mice.

Next, to examine whether HV influences Rspo3-induced improvement of glucose intolerance and insulin resistance in obese KK-Ay mice, we performed glucose tolerance and insulin tolerance tests. In SO mice, Rspo3 induction ameliorated glucose intolerance and insulin resistance in obese mice, but these improvements were partially blocked by HV (Fig 5K and 5L). These results suggested hepatic Rspo3 induction to ameliorate glucose intolerance and insulin resistance via the combination of decreased gluconeogenesis and increased glycolysis locally in the liver and/or improved peripheral insulin sensitivity systemically. While the former mechanisms do not require inter-organ communication mediated by the afferent vagus and efferent sympathetic nerves, the latter does.

## Suppression of Rspo3 leads to metabolic derangements promoting obesity development

Next, we examined whether suppression of hepatic Rspo3 exerted any effects on glucose metabolism and body composition. We generated tamoxifen-induced liver-specific Rspo3 knockout mice (iLiRspo3KO mice) by crossing serum albumin promoter-CreER mice and

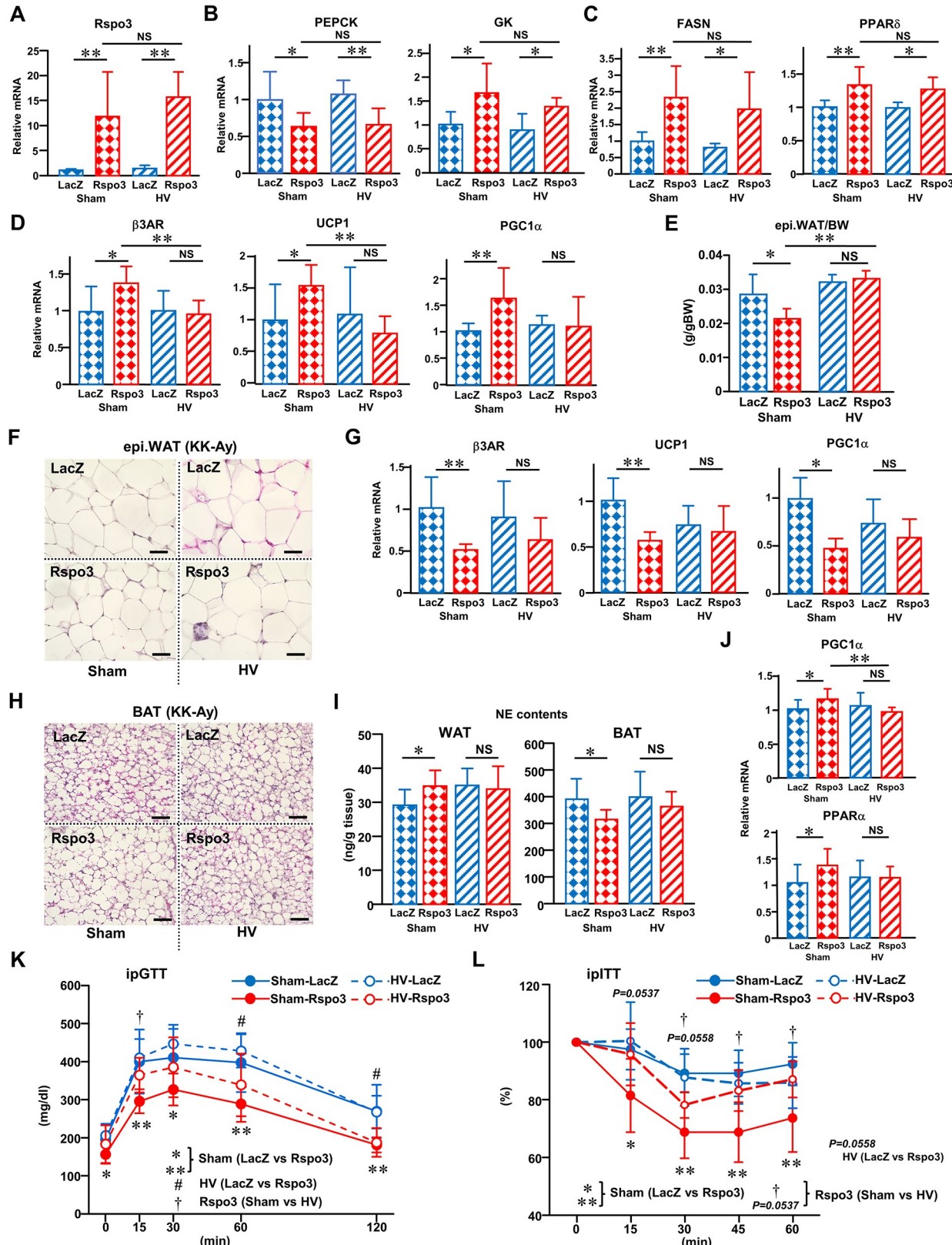

**Fig 5. A neuronal inter-organ communication system is involved in Rspo3-induced remote effects. (A–L)** Selective HV or a sham operation (Sham) was performed 7 days prior to the adenovirus administration. Rspo3 (red bars and circles) or LacZ (blue bars and circles) adenovirus was

administered to these mice. **(A)** Hepatic Rspo3 expression ($n$ = 5–8), **(B)** hepatic PEPCK and GK expressions ($n$ = 5–8), and **(C)** hepatic FASN and PPARδ expressions ($n$ = 5–8) were examined on day 7 after adenovirus administration. **(D)** β3AR, UCP1, and PGC1α mRNA expressions in epididymal WAT ($n$ = 6–7), **(E)** epididymal WAT weight ($n$ = 4–6), **(F)** representative histological analysis images of epididymal WAT (HE staining), **(G)** β3AR, UCP1, and PGC1α mRNA expressions in BAT (β3AR, UCP1; $n$ = 5–8, PGC1α; $n$ = 10–14), **(H)** representative histological analysis images of BAT (HE staining), **(I)** NE contents in WAT and BAT ($n$ = 6–7), and **(J)** PGC1α and PPARα mRNA expressions in skeletal muscle ($n$ = 7–11) were obtained 7 days after adenovirus administration. The scale bars indicate 50 μm. **(K)** Glucose tolerance and **(L)** insulin tolerance tests were both performed on day 7 after adenovirus administration ($n$ = 5–14). The data underlying the graphs shown in the figure can be found in **S1 Data**. Data are presented as means ± SD. *$P$ < 0.05, **$P$ < 0.01, #$P$ < 0.05, †$P$ < 0.05 by the unpaired $t$ test. b3AR, b3-adrenergic receptor; BAT, brown adipose tissue; FASN, fatty acid synthetase; GK, glucokinase; HE, hematoxylin eosin; HV, hepatic vagotomy; NE, norepinephrine; PEPCK, phosphoenolpyruvate carboxykinase; PPAR, peroxisome proliferator-activated receptor; Rspo3, R-spondin3; WAT, white adipose tissue.

Rspo3-floxed mice obtained employing CRISPR/CAS9 genome editing technology (S17 Fig). Tamoxifen administration at a low dose to these mice decreased hepatic Rspo3 expression by almost 20%, followed by a much greater degree of decrease in Rspo3 protein levels (by almost 48%) (Fig 6A). RNA-Scope revealed that Rspo3-positive signals are also decreased in iLiRspo3KO mice (by 13%) (S18A Fig) though to a milder degree than indicated by the results of quantitative RT-PCR (Fig 6A). Suppression of hepatic Rspo3 to this degree changed neither cumulative food intake nor body weight gain for the 1-week period from day 28 or 4, respectively, after tamoxifen administration (S18B Fig). Under these conditions, hepatic GK expression was locally decreased but hepatic PEPCK expression was unchanged in iLiRspo3KO mice (Fig 6B) with no change in liver weight (S18C Fig). In addition, while hepatic LGR5, a downstream target of the Rspo3/Wnt signaling pathway, as well as Oat, Rdh9, and Cyp7a1, markers of genes in the pericentral zone, were decreased, while hepatic Cyp2f2, a marker of genes in the periportal zone, tended to be increased (S18D and S18E Fig). Suppression of Rspo3 exacerbated glucose intolerance and decreased insulin sensitivity in normal chow-fed iLiRspo3KO mice (Fig 6C and 6D), with these non-beneficial effects persisting 2 months after tamoxifen administration (S18F Fig). The examination of insulin signaling demonstrated that suppression of Rspo3 down-regulated insulin-stimulated serine473 phosphorylation of Akt in remote organs, such as skeletal muscle and epididymal WAT, as well as in the livers of iLiRspo3KO mice (Fig 6E).

As for remote organs, UCP1 expression in WAT was decreased (Fig 6F) with no change in WAT weight (S18G Fig), and UCP1 expression in BAT was increased (Fig 6G). BCAA-catabolizing enzyme (BCKDH) in BAT was also increased in these iLiRspo3KO mice (Fig 6G). These observations are consistent with the results showing suppression of Rspo3 to increase $VO_2$ and to slightly increase RQ, leading to up-regulation of whole-body energy expenditure (Fig 6H), a phenomenon with consequences opposite those observed in obese Rspo3-mice (Fig 4H). We examined whether suppression of Rspo3 impacts NE-induced $VO_2$ change, and obtained results indicating that hepatic Rspo3 suppression increases the $VO_2$ elevation in response to NE treatment (Fig 6I). In addition, β2AR and PGC1α expressions in skeletal muscle were decreased, which in turn decreased skeletal muscle expression of BCKDH (Fig 6J). Suppression of hepatic Rspo3 down-regulates BCAA metabolism in skeletal muscle, thereby diminishing the effective utilization of BCAAs, which results in deterioration of the capacity to maintain skeletal muscle functions [34]. Skeletal muscle expression of PPARα was also decreased in these mice (Fig 6J), a phenomenon with consequences opposite those observed in obese Rspo3-mice (Fig 4K). Therefore, suppression of hepatic Rspo3 is associated with obesity-induced insulin resistance, systemically, as well as diabetes and changes in body compositions including the distributions of adipose tissue and skeletal muscle.

Next, tamoxifen administration at a moderate dose to these mice decreased hepatic Rspo3 expression, by almost 40% (S18H Fig), as well as the number of Rspo3-positive hepatocytes in

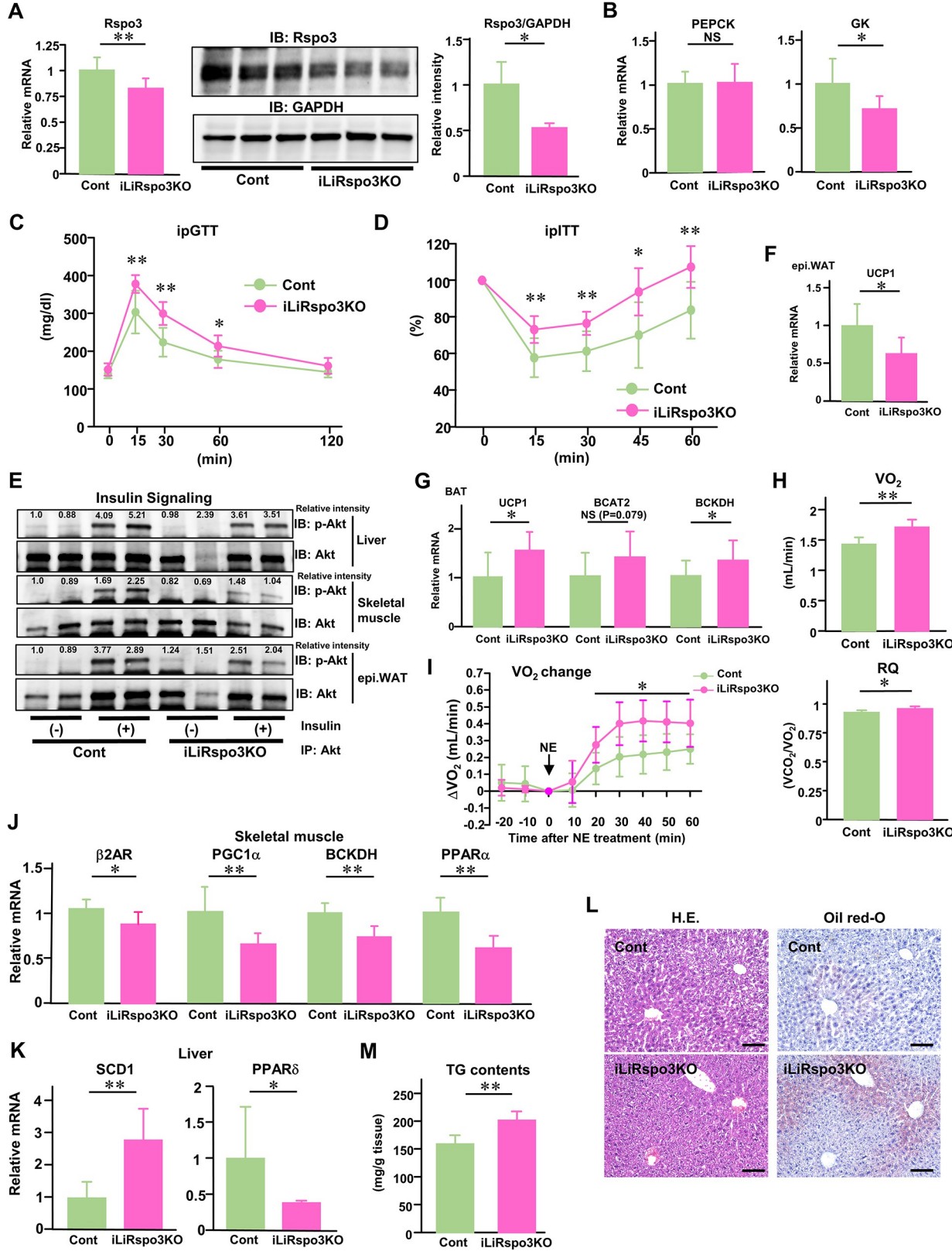

**Fig 6. Suppression of Rspo3 leads to metabolic derangements promoting obesity development. (A–M)** Data from iLiRspo3KO mice (pink bars and circles) and control mice (bright green bars and circles) are presented. **(A)** Hepatic Rspo3 mRNA expression ($n$ = 7–8) and

immunoblotting of liver extracts with anti-Rspo3 and anti-GAPDH antibodies and the relative intensities (the ratios of Rspo3/GAPDH protein volume), **(B)** hepatic PEPCK and GK mRNA expressions were examined on day 13 after tamoxifen (40 μg/g body weight) administration ($n = 7$–9). **(C)** Glucose tolerance and **(D)** insulin tolerance tests were performed on day 7 after tamoxifen (40 μg/g body weight) administration ($n = 7$–8). **(E)** Insulin-stimulated serine473 phosphorylation of Akt protein in the liver, skeletal muscle and WAT were immunoblotted with anti-phospho-Akt (Ser473) antibody, or just the antibody against Akt. The relative intensities (the ratios of p-Akt/Akt protein volume) were indicated in the p-Akt panels. **(F)** UCP1 mRNA expression in epididymal WAT ($n = 7$), **(G)** UCP1, BCAT2 and BCKDH mRNA expressions in BAT (UCP1; $n = 6$–7, BCAT2, BCKDH; $n = 11$–12), **(H)** VO2 and RQ ($n = 4$–7), **(I)** NE-induced VO$_2$ changes ($n = 4$–8), and **(J)** β2AR, PGC1α, BCKDH, and PPARα mRNA expressions in skeletal muscle were examined on day 13 after tamoxifen (40 μg/g body weight) administration (β2AR; $n = 14$–16, PGC1α, BCKDH, PPARα; $n = 7$–8). **(K)** Hepatic SCD1 and PPARδ mRNA expressions ($n = 6$–8), **(L)** representative histological analysis images of the liver, HE staining and Oil red-O staining, and **(M)** hepatic TG contents ($n = 7$–9) were examined on day 2 after tamoxifen (120 μg/g body weight) administration. The scale bars indicate 100 μm. The data underlying the graphs shown in the figure can be found in **S1 Data**. Data are presented as means ± SD. *$P < 0.05$, **$P < 0.01$ by the unpaired $t$ test. b2AR, b2-adrenergic receptor; BCAT2, branched-chain AA aminotransferase 2; BCKDH, branched-chain ketoacid dehydrogenase; GK, glucokinase; HE, hematoxylin eosin; NE, norepinephrine; PEPCK, phosphoenolpyruvate carboxykinase; PPAR, peroxisome proliferator-activated receptor; RQ, respiratory quotient; Rspo3, R-spondin3; SCD1, stearoyl CoA desaturase 1; TG, triglyceride; WAT, white adipose tissue.

the pericentral zone (S18I Fig). Under these conditions, while hepatic LGR5 and Axin2, markers of the downstream target of the Rspo3/Wnt signaling pathway, and Cyp7a1, a marker of genes in the pericentral zone, were decreased, hepatic Cyp2f2, a marker of genes in the periportal zone, was increased (S18J Fig). Suppression of Rspo3 increased hepatic SCD1 expression, while decreasing hepatic PPARδ expression (Fig 6K), a phenomenon with consequences opposite those observed in obese Rspo3-mice (Fig 3F and 3J). These results raise the possibility that the inter-organ communication system operating under the control of the hepatic PPARδ-to-muscle PPARα axis is negatively regulated, resulting in decreased fatty acid utilization in skeletal muscle. In addition, suppression of Rspo3 increased lipid droplets, as revealed by Oil red-O staining and calculated TG contents of the liver in normal chow-fed iLiRspo3KO mice, indicating the progression of fatty liver change (Fig 6L and 6M). These results demonstrate a reduction in hepatic Rspo3 expression of only 20% to 40% to exacerbate obesity-associated features, suggesting that Rspo3 may be a key player contributing to the regulation of both hepatic metabolic zonation and systemic glucose metabolism, as well as body composition, via inter-organ communication systems.

## Discussion

Metabolic homeostasis of nutrients, in mammals, is maintained via inter-organ communications, which are mediated by humoral secreted factors, the autonomic nervous system and the vascular system, constructing a myriad of networks linking remote organs in the body [38]. Several organs secrete an array of bioactive polypeptides, such as adipokines [39], hepatokines [40], and myokines [41], into the circulation, contributing to maintenance of systemic metabolism. As with these humoral factors, global research attention is now being focused on the roles of neuronal communication systems in this inter-tissue metabolic coordination [42–45]. As for the significance of the blood vascular system [46], suppression of endothelial inflammation or ER stress improves obesity-induced insulin resistance, leading to lifespan prolongation [47,48]. Therefore, most organs in the body function as metabolic sensors for a variety of nutrients and as production sites for many of the important signals, regarding nutritional status, which are transmitted to other organs via humoral factors, neuronal or vascular systems.

Conducting studies under the broad umbrella of global basic sciences, we have identified an inter-organ communication system mediated by a neuronal network originating in the liver and have proposed that this system plays an important role in the maintenance of homeostatic metabolism [38]. In detail, neuronal signals from the liver were recently identified as regulating basal metabolic rates [37], increasing pancreatic β cell mass [49,50], and contributing to obesity-related hypertension [51]. However, in the present age of satiation, obesity develops

and disturbs this metabolic homeostasis, resulting in pathophysiological states such as diabetes and metabolic syndrome [52], which are among the highest priority problems, in terms of social and medical economics, faced by the modern world.

In this study, we focused on the spatiotemporal sensing and interacting mechanism(s) of metabolic zonation in the liver. Hepatic landmark genes of metabolism for the different lobular zones were previously determined by both laser microdissection [53] and spatiotemporal transcriptomics studies [54–56]. Thus, metabolic features, under the control of these spatially restricted gene expressions, are considered to be positional markers of the different zones. As for the mechanism(s) underlying this metabolic zonation, the Wnt/β-catenin signaling pathway is a master regulator for gene expressions of the pericentral metabolic enzymes according to the results of studies showing a pericentral-portal gradient in the activity of this signaling pathway [11,57]. This activity gradient is formed by both Wnt and Rspo ligands expressed around central veins and sinusoidal endothelial cells [13,14,58]. However, the precise mechanisms mediating the zonated expression pattern of these ligands and constructing the activity gradient of their downstream signaling pathway has yet to be clarified.

We next investigated the role of Rspo3, a key molecule modulating hepatic zonation via the Wnt/β-catenin signaling pathway [4], by regulating its expression throughout the liver, resulting in the identification of a new inter-organ "neuronal" communication system functioning to maintain metabolic homeostasis. Furthermore, we have strived to elucidate both the pathophysiological significance and the mechanism(s) by which this inter-organ communication system is disturbed during obesity development, thereby contributing to the exacerbation of obesity-associated features.

First, hepatic Rspo3 expression was shown to be enhanced and to converge around central veins in response to feeding as compared to the fasting state (i.e., spatiotemporal characteristics of Rspo3), while being attenuated in obesity, reflecting the physiological and pathophysiological significance of Rspo3.

Second, hepatic Rspo3 induction ameliorates obesity-associated features, such as fatty liver change, diabetes, and insulin resistance including those impacting remote organs. Locally in the liver, hepatic Rspo3 induction, reconstituting metabolic zonation, revealed a novel inter-nutrient mechanism by which the balance between lipid and glucose metabolism is maintained, thereby playing an important role in suppressing obesity-related fatty liver progression (S12C Fig). Outside the liver, hepatic Rspo3 induction ameliorated reduced skeletal muscle quality in obesity by restoring insulin signaling and up-regulating muscle PPARα expression and muscle fatty acid utilization. These remote effects of Rspo3 on skeletal lipid utilization through PPARα suggest the possible involvement of the neuronal communication operating under the hepatic vagus as well as non-neuronal communication operating under the hepatic PPARδ-phosphatidylcholine-muscle PPARα axis [22]. This complicated inter-organ system has been considered to play a significant role in maintaining metabolic homeostasis mainly in the postprandial state, which is compatible with the finding that hepatic Rspo3 converges in the pericentral zones after feeding and exerts several metabolic functions, as described in the present report. In addition, hepatic Rspo3 induction diminished enlargement of WAT in obese mice, while it restored sustained stimulation of BAT-mediated adaptive thermogenesis against obesity development. These remote effects on different adipose tissues were almost completely nullified by selective dissection of the hepatic vagal nerve. The remote effect in WAT is similar to the phenomenon mediated by neuronal signals originating in hepatic PPARγ expression, which modifies white adipocyte characteristics and regulates energy metabolism [37]. However, the finding that hepatic PPARγ expression was changed in lean Rspo3-mice but not necessarily in obese Rspo3-mice suggests the existence of a neuronal mechanism, different from that originating in hepatic PPARγ, which functions in the

maintenance of WAT metabolism. The remote effect in BAT is similar to the so-called energy-saving machinery mediated by neuronal signals originating in hepatic GK expression, down-regulating thermogenesis-related genes in BAT, i.e., BAT thermogenic physiology [59]. On the contrary, as observed in the setting of obesity, decreases in hepatic Rspo3 expression and its function contribute to reducing the stimulation signal delivered to WAT-derived thermogenesis, resulting in WAT enlargement, while it contributes to producing that delivered to BAT-derived adaptive thermogenesis against obesity development, a phenomenon which precisely reflects the pathophysiology of obesity. As previously reported, BAT-enriched adipokine (Adissp) serves as a major upstream signaling component especially in inguinal WAT thermogenesis, but not either BAT or epididymal WAT thermogenesis, and improves glucose homeostasis, suggesting the significance of a mechanism that distinguishes and balances thermogenesis derived from different regions of adipose tissues [60]. Our present results raise the possibility that reconstitution of hepatic metabolic zonation by Rspo3 induction in obesity, with increased WAT-derived thermogenesis and reversal of overstimulated BAT-derived thermogenesis, may lead to balancing and redistribution of thermogenesis derived from different adipose tissues such as WAT and BAT. Thus, interpretation of the complex results obtained in this study points to the involvement of a newly-identified "neuronal" inter-organ communication system arising from the control of hepatic metabolic zonation by Rspo3 in the maintenance of thermogenic homeostasis, which differs between adipose tissues such as WAT and BAT, highlighting the novel and diverse roles of neuronal signals transmitted by the hepatic vagal nerve.

Third, selective knockdown of hepatic Rspo3 exacerbates diabetes and fatty liver change, and then decreases insulin sensitivity in remote organs such as skeletal muscle and adipose tissue as well as in the liver. Again, down-regulation of hepatic Rspo3 changes the expressions of genes relevant to adipose tissue and skeletal muscle, i.e., genes related to thermogenesis as well as those related to myogenesis, mitochondrial biogenesis, and lipid utilization, respectively, leading to deterioration of body composition and organ quality.

Taking the present results into consideration, the Rspo3-induced reconstitution of hepatic metabolic zonation, which is disturbed in obesity, plays an important role in regulating local hepatic and systemic metabolism of nutrients and may restore the systemic composition of adipose tissues while improving skeletal muscle quality, by producing a newly identified complicated inter-organ communication system. On the other hand, prolonged obesity disturbs the physiological and metabolic functions originating in the control of hepatic metabolic zonation by Rspo3, leading to the progression of obesity-associated features such as glucose intolerance, insulin resistance, fatty liver, and deterioration of organ quality (Figs 7 and S19).

A recent study, focusing on liver regeneration, implicated the proliferation signals from hepatocytes in the mid-lobular zones in the repopulation and regeneration of hepatocytes in the neighboring pericentral and periportal zones [61]. Namely, hepatocytes in these different lobular zones, which have distinct characteristics in liver regeneration [62] or metabolism (described in the present study), interact to construct novel inter-cellular or inter-organ communication systems, respectively, contributing to the maintenance of both liver and whole-body homeostasis.

However, a limitation of this study is that we did not strictly examine whether hepatic Rspo3 induction in obesity has effects more prolonged than those observed up to 23 days after adenovirus administration. The detailed mechanism, by which modulation of hepatic Rspo3 expression exerts long-term effects on whole-body metabolism and body composition, via an inter-organ communication system, awaits elucidation and merits further investigation.

In the currently aging society, with rising rates of diabetes, metabolic syndrome and sarcopenia, medical interventions designed to "extend healthy life expectancy" are urgently needed

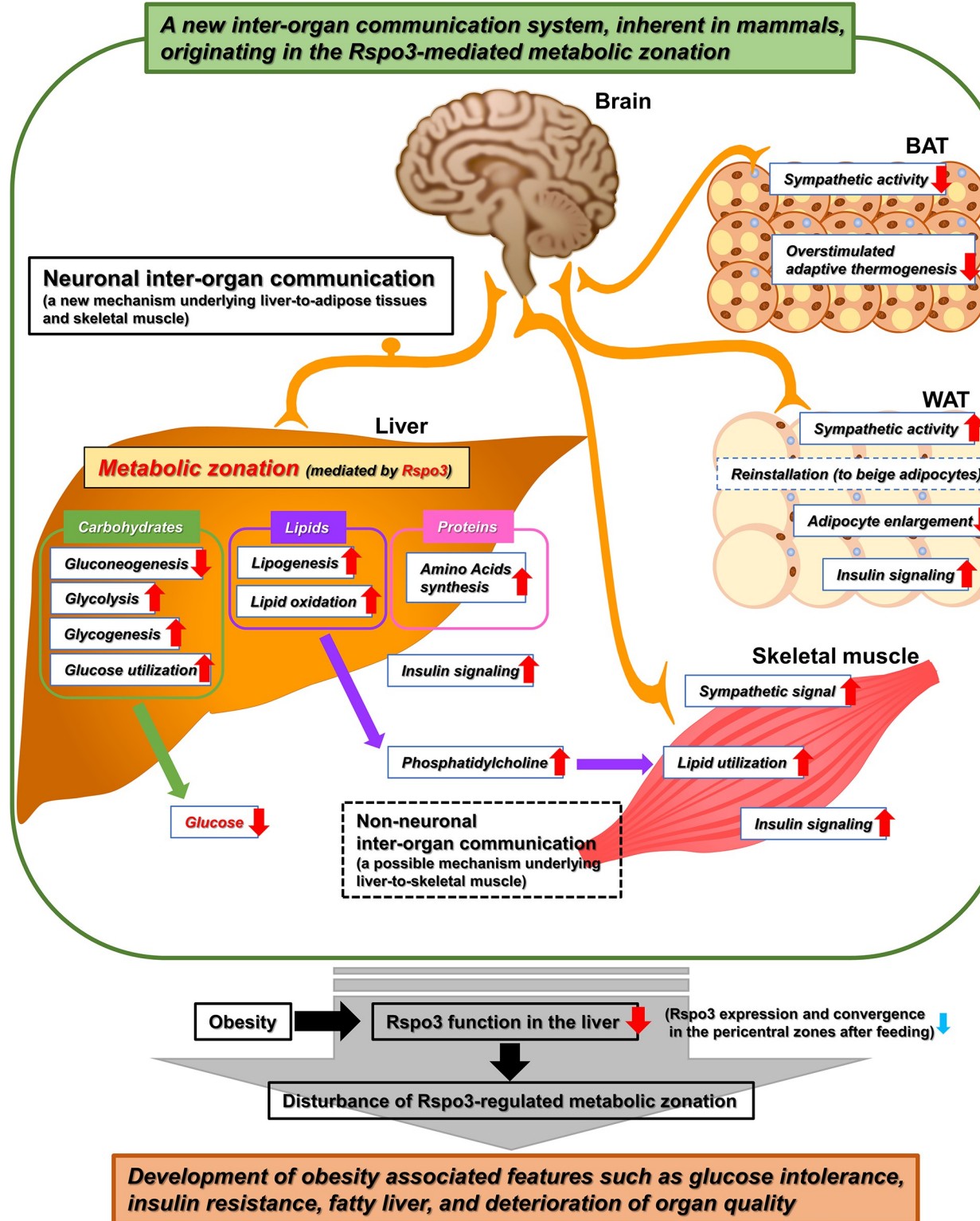

**Fig 7. The scheme of a newly identified inter-organ communication system.** Hepatic metabolic zonation mediated by Rspo3 plays an important role in maintaining metabolism in remote organs, such as adipose tissues and skeletal muscles, as well as locally in the liver through the inter-organ coordination consisting of neuronal and non-neuronal communication. However, in the setting of obesity, the impaired Rspo3 function in the liver leads to the disturbance of this Rspo3-mediated physiological inter-organ mechanism, paradoxically resulting in progression of pathophysiological features such as glucose intolerance, insulin resistance, fatty liver, and deterioration of organ quality. BAT, brown adipose tissue; Rspo3, R-spondin3; WAT, white adipose tissue.

[63]. Recently, new technical tools targeting neuronal inter-organ communication have been developed. The first involves stimulating the hepatoportal nerve plexus with focused ultrasound to restore glucose homeostasis in diabetic animals [64]. The second innovation is stimulating parasympathetic nerves to pancreatic β cells with an optogenetic signal to increase insulin secretion, a strategy applicable to diabetic patients [65]. If an innovative technology to selectively stimulate hepatic Rspo3 or its interacting vagal nerves can be developed, it may open novel avenues to accessing this newly identified inter-organ communication system derived from modulation of hepatic Rspo3-mediated metabolic zonation. Therefore, targeting Rspo3 is very important for achieving therapeutic innovations and applications for patients suffering from diabetes, obesity, and obesity-associated diseases. Such medical breakthroughs are eagerly awaited by the public.

## Materials and methods

### Animals

Animal studies were conducted in accordance with Teikyo University institutional guidelines. All experimental protocols had been approved by the Institutional Animal Care and Use Committee of the Teikyo University Environmental & Safety Committee (Teikyo University Ethics Committee for Animal Experiments) and assigned numbers as 18–037, 18–038, and 18–040. Male C57BL/6J and C57BL/6J-Lep$^{ob/ob}$ (ob/ob) mice were purchased from Japan SLC, Inc. (Shizuoka, Japan). C57BL/6J-DIO mice were obtained by 4-week feeding of a high-fat diet (HFD-60, with total calories provided by 62.2% lipids, 19.6% carbohydrates, and 18.2% proteins) (Oriental Yeast Co., Ltd., Tokyo, Japan) beginning at 5 weeks of age. Male KK-Ay mice were purchased from CLEA Japan, Inc. (Tokyo, Japan). Rspo3 floxed mice (Rspo3$^{flox/flox}$) with a C57BL/6 background were created by means of CRISPR/CAS9 technology at the Institute of Laboratory Animals, Tokyo Women's Medical University (Tokyo, Japan), as previously described [66,67]. The CAS9 endonuclease used in this study is EnGen Cas9 NLS, S. pyogenes (#M0646T, New England Biolabs, Ipswich, Massachusetts, United States of America). The CRISPR RNA (crRNA), trans-activating CRISPR RNA (tracrRNA), and single-strand oligo donor DNA (ssODN) were created by Fasmac Co., Ltd. (Kanagawa, Japan). These sequences are described in **S1 Text**. The loxP sites were inserted both up- and down-stream from the first exon of the Rspo3 genome using CRISPR/CAS9. The serum albumin promoter-CreERT2 (SA-CreER) mice were obtained from Tohoku University (Miyagi, Japan) [68]. To obtain tamoxifen-inducible liver-specific Rspo3 knockout (iLiRspo3KO) mice, we crossed SA-CreER mice and Rspo3$^{flox/flox}$ mice. At 9 weeks of age, male SA-CreER; Rspo3$^{flox/flox}$ mice and male Rspo3$^{flox/flox}$ mice, as controls, were injected intraperitoneally with tamoxifen (#T5648, Sigma, St. Louis, Missouri, USA), at the low dose of 40 μg/g body weight, dissolved in corn oil (#C8267, Sigma) every 24 h for 4 consecutive days (total dose of 160 μg/g body weight). These mice were analyzed 7 and 13 days after 4 tamoxifen administrations. At the moderate dose of 120 μg/g body weight, tamoxifen was injected intraperitoneally to 6-week-old male SA-CreER; Rspo3$^{flox/flox}$ mice and control mice every 24 h for 3 consecutive days (total dose of 360 μg/g body weight). These mice were analyzed 2 days after 3 tamoxifen administrations. All of these animals were housed in an air-conditioned environment, with a 12-h light-dark cycle. Only male mice were used for these experiments.

### Preparation of recombinant adenoviruses

Recombinant adenovirus, containing the murine R-spondin3 (Rspo3) under the control of the CMV promoter, was prepared using the Adeno-X Adenoviral System 3 (#632267, Takara Bio Inc., Otsu, Japan). Rspo3 fragments for constructing the adenovector are described in **S1 Text**.

Recombinant adenovirus containing the bacterial β-galactosidase gene was used as a control [69]. Recombinant adenoviruses were propagated in HEK293 cells and purified employing cesium chloride gradient ultracentrifugation [70]. Titers of these viruses were determined by endpoint cytopathic effect assay.

### Administration of recombinant adenoviruses

Recombinant adenoviruses were injected into the tail veins of either standard chow-fed 8-week-old male C57BL/6J, ob/ob, and KK-Ay mice or high fat-fed 8-week-old DIO mice at a dose of $5.0 \times 10^7$ plaque forming units as previously reported [37].

### Immunoblotting

Tissue samples were homogenized in ice-cold RIPA Lysis buffer containing 1×RIPA Lysis buffer, 2 mM phenylmethylsulfonyl fluoride, 1 mM sodium orthovanadate, and protease inhibitor cocktail at a 1:100 dilution (#sc-24948A, Santa Cruz Biotechnology, Inc., Santa Cruz, California, USA). Tissue homogenates were centrifuged and the supernatants including tissue protein extracts (20 µg total protein) were boiled in Laemmli buffer containing 10 mM dithiothreitol, then subjected to SDS-polyacrylamide gel electrophoresis (SDS-PAGE) employing the Trans-Blot Turbo Transfer System (Bio-Rad Laboratories, Inc., Hercules, California, USA). The liver samples were immunoblotted with antibodies including anti-Rspo3 antibody (#32155, GeneTex, Inc., Alton Pkwy Irvine, California, USA), anti-PDK4 antibody (#12949, Proteintech Group, Inc., Rosemont, Illinois, USA), anti-PPARδ antibody (#74076, Cell Signaling Technology, Inc., Danvers, Massachusetts, USA), anti-Cpt1a antibody (#114337, GeneTex, Inc.), and anti-GAPDH antibody (#5174, Cell Signaling Technology, Inc.), as previously reported. The immunoblots using antibodies obtained from GeneTex, Inc., Proteintech Group, Inc., and Cell Signaling Technology, Inc. were routinely performed at a 1:2,000 dilution in Can Get Signal (#NKB-101, TOYOBO, Osaka, Japan), followed by secondary antibodies (ECL Anti-Rabbit IgG, horseradish peroxidase (HRP)-linked Whole Antibody (from donkey)) (#NA934V, GE Healthcare, Buckinghamshire, United Kingdom) at a 1:5,000 dilution. The immunoblots were visualized with an ECL Prime Western Blotting System (#GERPN2232, Merck KGaA, Darmstadt, Germany).

### Immunoblotting of nuclei

The nuclear fraction was isolated as described in the protocol of Schreiber and colleagues [71] with slight modifications. Tissue samples were homogenized in ice-cold buffer (pH 7.9) containing 10 mM HEPES, 1.5 mM MgCl2, 10 mM KCL, 0.5 mM DTT, 0.05% NP-40, and protease inhibitor (cOmplete ULTRA, mini, EASYpack (#05892970001, Merck KGaA)). Tissue homogenates were centrifuged and the pellets including nuclear protein extracts were dissolved in buffer containing 5 mM HEPES, 1.5 mM MgCl2, 0.2 mM EDTA, 0.5 mM DTT, 26% glycerol, and 0.3 M NaCl. Dissolved pellets were centrifuged and the supernatants were boiled in Laemmli buffer containing 10 mM dithiothreitol, then subjected to SDS-PAGE employing the Trans-Blot Turbo Transfer System (Bio-Rad Laboratories, Inc.). The nuclear samples from the liver were immunoblotted with antibodies including anti-Foxo1 antibody (#18592-1-AP, Proteintech Group, Inc.) and anti-HDAC1 antibody (#2062, Cell Signaling Technology, Inc.), as previously reported. The immunoblots using antibodies obtained from Proteintech Group, Inc. and Cell Signaling Technology were routinely performed at a 1:2,000 dilution with Can Get Signal (TOYOBO), followed by secondary antibodies (ECL Anti-Rabbit IgG, HRP-linked Whole Antibody) (#NA934V, GE Healthcare) at a 1:5,000 dilution. The immunoblots were visualized with an ECL Prime Western Blotting System (Merck KGaA).

## Blood analysis

Blood glucose levels were measured using the glucose analyzer GLUTESTMINT (SANWA KAGAKU KENKYUSHO CO., LTD., Nagoya, Japan) and serum insulin levels were assayed with the Insulin ELISA Kit (#M1102, Morinaga Institute of Biological Science, Inc., Kanagawa, Japan). Blood Rspo3 levels were measured with the R-Spondin-3 PicoKine ELISA Kit (#EK1525, Boster Biological Technology, Pleasanton, California, USA). Blood phosphatidylcholine levels were measured with the Phosphatidylcholine Assay Kit (#Ab83377, Abcam plc., Cambridge, UK).

## Histological analysis

Liver tissue, WAT, and BAT were removed and fixed in 10% formalin and embedded in OCT compounds, then subjected to the bio-tissue freezing slicing method (Leica CM1520 cryostat, Leica Biosystems, Nussloch, Germany). Tissue sections were stained with HE, Oil red-O, and PAS. The images of these stained samples were obtained with an all-in-one Fluorescence Microscope (BZ-X800, KEYENCE, Osaka, Japan) and analyzed using the BZ-X Analyzer software (KEYENCE).

## Fluorescent immunostaining analysis of the liver

For formalin fixation, the liver was removed and fixed in 10% formalin. For reperfusion fixation, mice anesthetized with medetomidine (0.3 mg/kg), midazolam (4 mg/kg), and butorphanol tartrate (5 mg/kg) were perfused transcardially with 4% paraformaldehyde (PFA), and then the liver was removed and fixed in 4% PFA. These liver samples were embedded in OCT compounds, then subjected to the bio-tissue freezing slicing method (Leica Biosystems). Five-μm-thick sections were immunostained with anti-Rspo3 polyclonal rabbit antibody (#32155, Gene-Tex, Inc.) at a 1:100 dilution as the primary antibody and Cy3 affinity pure donkey anti-rabbit IgG (#711-165-152, Jackson ImmunoResearch Laboratories, Inc., West Grove, Pennsylvania, USA) at a 1:200 dilution as the secondary antibody. In addition, liver sections were immunostained with anti-HNF4α polyclonal rabbit antibody (#bs-3828R, BIOSS Inc., Woburn, Massachusetts, USA), and anti-CK19 polyclonal rabbit antibody (#14965-1-AP, Proteintech Group, Inc.) at a 1:100 dilution [72], and anti-Ki67 rabbit monoclonal antibody (#12202S, Cell Signaling Technology, Inc.) at a 1:200 dilution as the primary antibodies and Cy3 affinity pure donkey anti-rabbit IgG (#711-165-152, Jackson ImmunoResearch Laboratories, Inc.) at a 1:200 dilution as the secondary antibody. Liver sections were immunostained with anti-CD31 rat antibody (#550274, BD Bioscience, Franklin Lakes, New Jersey, USA) at a 1:20 dilution as the primary antibody and Goat Anti-Rat IgG (Cy5) (#ab6565, Abcam plc.) at a 1:100 dilution as the secondary antibody. The nuclei of hepatocytes were visualized with Cellstain DAPI solution (#D523, DOJINDO LABORATORIES, Kumamoto, Japan). The images of these immunostained samples were obtained with an inverted confocal laser scanning microscope (Zeiss Super-Resolution Microscope LSM880 with AiryScan, Carl Zeiss, Oberkochen, Germany) and analyzed using the ZEISS ZEN and Imaris software (Carl Zeiss). The intensity of Rspo3 immunofluorescence was measured on 4 lines, each 20 μm from blood vessel walls along the lines at right angles centered on the hepatic central veins obtained at random, according to the manufacturer's instructions.

## Fluorescent immunostaining analysis of rRPa of the brain

For formalin fixation, the brain was removed and fixed in 10% formalin. These brain samples were embedded in OCT compounds, then subjected to the bio-tissue freezing slicing method (Leica Biosystems). Ten-μm-thick sections were immunostained with anti-c-fos monoclonal

rabbit antibody (#2250S, Cell Signaling Technology, Inc.) at a 1:500 dilution as the primary antibody and Cy3 affinity pure donkey anti-rabbit IgG (#711-165-152, Jackson ImmunoResearch Laboratories, Inc.) at a 1:200 dilution as the secondary antibody. The images of these immunostained samples were obtained with an all-in-one Fluorescence Microscope (BZ-X800, KEYENCE) and analyzed using BZ-X Analyzer software (KEYENCE).

### RNA-Scope (in situ hybridization)

The liver was removed and fixed in 10% formalin and embedded in OCT compounds, then subjected to the bio-tissue freezing slicing method (Leica Biosystems). Five-μm-thick liver sections were subjected to the hybridization of RNAscope-specific probes. Three probes against mouse Rspo3 (RNAscope Probe-Mm-Rspo3, #402011), mouse Ppib (RNAscope Positive Control Probe-Mm-Ppib, #313911), and DapB (RNAscope Negative Control Probe-DapB, #310043) were prepared employing Advanced Cell Diagnostics, Inc. (Newark, California, USA). We used the single-plex RNAscope assay according to the User Manual of RNAscope 2.5 HD Reagent Kit-Brown (#322300), in which Rspo3-positive signals (as dots) are detected with DAB [73]. Quantification of Rspo3-positive signals in the liver sections is provided by counting the numbers of dots according to the manufacturer's instructions for the BZ-X Analyzer software (KEYENCE).

### Glucose and insulin tolerance tests

Glucose and insulin tolerance tests were performed 5 to 20 days after adenovirus administration or 7 days to 2 months after 4 tamoxifen administrations, as previously described [37]. Insulin (0.15 U/kg of body weight for lean mice and 0.25 U/kg of body weight for obese and SA-CreER; Rspo3$^{flox/flox}$ mice) was injected into the intraperitoneal space.

### Immunoprecipitation

Mice fasted 16 h were injected with 200 μl of normal saline, with or without 10 U/kg body weight of insulin, via the tail vein. The liver, hindlimb muscle, and epididymal WAT were removed 300 s later and then homogenized in ice-cold RIPA Lysis buffer containing 1×RIPA Lysis buffer, 2 mM phenylmethylsulfonyl fluoride, 1 mM sodium orthovanadate, and a protease inhibitor cocktail at a 1:100 dilution (#sc-24948A, Santa Cruz Biotechnology, Inc.). The tissue homogenates were centrifuged and the supernatants including tissue proteins were used for immunoprecipitation with 5 μl of anti-Akt antibody (#9272, Cell Signaling Technology, Inc.) or anti-insulin receptor antibody (#3025, Cell Signaling Technology, Inc.), coupled with nProtein A-Sepharose 4 Fast Flow (#GE17-5280-01, Merck KGaA). Next, the samples immunoprecipitated with anti-Akt antibody were subjected to SDS-PAGE and then immunoblotted using anti-phospho-Akt (Ser473) antibody (#4060, Cell Signaling Technology, Inc.) or anti-Akt antibody at a 1:3,000 dilution in Can Get Signal (TOYOBO), followed by secondary antibodies (ECL Anti-Rabbit IgG, HRP-linked Whole Antibody) (#NA934V, GE Healthcare) at a 1:10,000 dilution, as previously described [37]. The samples immunoprecipitated with anti-insulin receptor antibody were immunoblotted using anti-phospho-Tyrosine antibody (#9411, Cell Signaling Technology, Inc.) or anti-insulin receptor antibody at a 1:3,000 dilution, as described above.

### Hepatic glycogen content

Frozen livers were homogenized, and glycogen contents were measured using a Glycogen Colorimetric/Fluorometric Assay Kit (#646–100, BioVision, Inc., Milpitas, California, USA), as previously described [74].

### Hepatic TG content

Hepatic TG contents were measured as previously described [37]. Frozen livers were homogenized in the buffer (137 mmol/L NaCl, 20 mmol/L Tris, pH 7.5, 1 mmol/L $MgCl_2$, 1 mmol/L $CaCl_2$, and 10% glycerol), followed by extraction with chloroform: methanol (2:1, vol:vol), drying, and resuspension in isopropanol. TG contents were measured using a LabAssay Triglyceride (#632–50991, FUJIFILM Wako, Osaka, Japan).

### Adipose NE contents

Adipose tissues were rapidly removed and weighed. Tissue samples were homogenized in 0.4 M perchloric acid and were centrifuged. Supernatant norepinephrine levels were measured commercially by SRL (Tokyo, Japan) [75].

### NE turnover

NE turnover was measured on the basis of the decrease in adipose tissue NE contents after the inhibition of catecholamine biosynthesis with α-methyl-D,L-p-tyrosine, methyl ester, hydrochloride (α-MT) (#sc-219470, Santa Cruz Biotechnology, Inc.). At 0 and 4 h after α-MT injection (200 mg/kg, i.p.), the mice were sacrificed by cervical dislocation and adipose tissues were rapidly removed and weighed. Tissue samples were homogenized in 0.4 M perchloric acid and then centrifuged. Supernatant NE levels were measured commercially by SRL (Tokyo, Japan) [75].

### Whole-body oxygen consumption and respiratory quotient (RQ)

Whole-body oxygen consumption and RQ were measured by an indirect calorimetric method using the Oxymax system (Equal 4 Chamber/Small Subject System), equipped with a four-chamber airtight metabolic cage, and the data obtained with this system were analyzed using CLAMS software (Columbus Instruments, Columbus, Ohio, USA). Mice, in non-anesthetized states, were acclimated to the individual metabolic cages for 1 day prior to the experiment. Volumes of oxygen consumed ($VO_2$) and carbon dioxide produced ($VCO_2$) for each metabolic cage were collected every 10 min for 2 days, with room air as a reference. The RQ was calculated by dividing the $VCO_2$ by the $VO_2$, used to estimate the contribution of fats and carbohydrates to whole-body energy metabolism in vivo.

### NE-induced whole-body oxygen consumption

For measurement of whole-body $O_2$ consumption after stimulation with NE, the mice were anesthetized with medetomidine, midazolam, and butorphanol tartrate, and $VO_2$ measurements were conducted for 30 min to obtain basal values. Mice were then briefly removed from the calorimetry chamber, injected intraperitoneally with 1 mg/kg body weight of L-norepinephrine hydrochloride (#74480, Merck KGaA), and returned to the chamber, followed by $VO_2$ measurement for another 60 min.

### Quantitative RT-PCR–based gene expressions

Total RNAs from the liver, WAT and BAT, and skeletal muscle were purified with miRNeasy Mini Kits (#217004, QIAGEN, Valencia, California, USA). cDNAs were synthesized from total RNA of several tissues with both RNasin Ribonuclease inhibitor (#N2111, Promega Corporation, Madison, Wisconsin, USA) and M-MLV-Reverse-Transcriptase-RNase-H-Minus (#M5301, Promega Corporation). Quantitative RT-PCR was performed by Taqman analysis (Thermo Fisher Scientific, Waltham, Massachusetts, USA) with a real time PCR quantitative system or a Syber Green System, THUNDERBIRD SYBR qPCR Mix (#QPS-201, TOYOBO

CO., LTD., Osaka, Japan) with MyGo Pro Real Time PCR (IT-IS Life Science Ltd, Joyce House, Ireland). The relative amount of mRNA was calculated with GAPDH mRNA as the invariant control [76]. The Taqman probes and PCR primers used in this study are presented in **S1 Text**.

### RNA-sequencing of hepatic tissues

RNA-sequence analyses were performed commercially by Cell Innovator Co., Ltd. (Fukuoka, Japan) [77]. GO term analysis was performed by using the web-accessible program DAVID (Database for Annotation, Visualization, and Integrated Discovery), which integrates functional genomic annotations. KEGG pathway analysis was also performed employing the DAVID program.

### Metabolomics of hepatic tissues

Metabolomic analysis was performed commercially by Human Metabolome Technologies, Inc. (Yamagata, Japan) [78].

### Hepatic vagotomy

Selective HV was performed as previously described [52]. After being anesthetized with an intraperitoneal injection containing a mixture of medetomidine (0.3 mg/kg), midazolam (4 mg/kg), and butorphanol tartrate (5 mg/kg), a laparotomy incision was made on the ventral midline and the abdominal muscle wall was opened with a second incision. The gastrohepatic ligament was severed using fine forceps, and the stomach was gently retracted, revealing the descending ventral esophagus and the ventral subdiaphragmatic vagal trunk. The hepatic branch of this vagal trunk was then transected using fine forceps or surgical scissors.

### Quantification and statistical analysis

The individual numerical values underlying the graphs and heatmaps shown in the figures can be found in **S1 Data**. The representative results derived from multiple experiments were displayed. All data are expressed as means ± SD. The statistical significance of differences was assessed employing the unpaired $t$ test.

## Supporting information

**S1 Fig. Azan-staining and immunofluorescence analysis.** Representative image of Azan-staining and representative immunofluorescence analysis image with anti-Rspo3 antibody and DAPI staining, of pericentral and periportal zones in the liver. The scale bars indicate 100 μm. (The scale bar in whole picture of the liver specimen indicates 1 mm.)
(TIF)

**S2 Fig. Immunofluorescence analysis of lean and obese mice.** Representative immunofluorescence analysis image with anti-Rspo3 antibody and DAPI staining of pericentral and periportal zones in the liver. The scale bars indicate 100 μm.
(TIF)

**S3 Fig. Hepatic gene profiles induced by Rspo3 in lean mice. (A–H)** Rspo3 (red bars and circles) or LacZ (blue bars and circles) adenovirus was administered to standard diet-fed control mice. **(A)** Rspo3 mRNA expression in WAT, BAT, and skeletal muscle were examined on day 7 after adenovirus administration (WAT, skeletal muscle; $n$ = 5, BAT; $n$ = 9–10). **(B)** Hepatic LGR5 and Axin2 mRNA expressions ($n$ = 6–7), **(C)** hepatic Oat, Cyp7a1, and Rdh9 mRNA

expressions ($n = 7$), **(D)** hepatic Hsd17 and Cyp2f2 mRNA expressions ($n = 6–7$) were examined on day 7 after adenovirus administration. **(E)** Hepatic Ki67 mRNA expressions were examined on day 7 ($n = 7$) and day 23 ($n = 5$) after adenovirus administration. **(F)** Hepatic gene expressions associated with glucose and lipid metabolism were examined on day 23 ($n = 5$) after adenovirus administration. **(G)** Glucose tolerance and **(H)** insulin tolerance ($n = 5$) tests were performed on day 20 and day 19 after adenovirus administration, respectively. The data underlying the graphs shown in the figure can be found in **S1 Data**. Data are presented as means ± SD. *$P < 0.05$, **$P < 0.01$ by the unpaired $t$ test.
(TIF)

**S4 Fig. Hepatic gene profiles induced by Rspo3 in obese mice. (A–I)** Rspo3 (red bars) or LacZ (blue bars) adenovirus was administered to standard diet-fed ob/ob mice. **(A)** Rspo3 mRNA expression in WAT, BAT, and skeletal muscle were examined on day 7 after adenovirus administration ($n = 10–12$). **(B)** Representative RNA-Scope analysis images of the liver in obese Rspo3 mice with specific Rspo3 probe. The numbers of Rspo3-positive signals (dots) were measured randomly (2 areas per mice, $n = 3$). **(C)** Circulating Rspo3 levels were measured on day 3 ($n = 6$) and 7 (fast, $n = 13–14$, fed, $n = 7$) after adenovirus administration. **(D)** Hepatic LGR5 and Axin2 mRNA expressions ($n = 6$); **(E)** hepatic Oat, Cyp7a1, and Rdh9 mRNA expressions ($n = 5–6$); **(F)** hepatic Hsd17 and Cyp2f2 mRNA expressions ($n = 6$) were examined on day 7 after adenovirus administration. **(G)** Representative immunofluorescence analysis images of the liver in obese Rspo3 mice with anti-Rspo3 antibody and DAPI staining, followed by fluorescence analysis of ZsGreen1, and **(H)** representative immunofluorescence analysis images of the liver in obese Rspo3 mice with anti-Ki67 antibody and DAPI staining, followed by fluorescence analysis of ZsGreen1, and **(I)** hepatic Ki67 mRNA expressions ($n = 4–5$) were examined on day 23 after adenovirus administration. The scale bars indicate 100 μm. The data underlying the graphs shown in the figure can be found in **S1 Data**. Data are presented as means ± SD. **$P < 0.01$ by the unpaired $t$ test.
(TIF)

**S5 Fig. Gene ontology (GO) analysis of RNA-sequence of the liver.** RNA-sequence analysis was performed by using mRNA sample of the liver from lean mice and obese (ob/ob) mice ($n = 4$), followed by GO analysis. The data underlying the graphs shown in the figure can be found in **S1 Data**.
(TIF)

**S6 Fig. KEGG pathway analysis of RNA-sequence of the liver.** RNA-sequence analysis was performed by using mRNA sample of the liver from lean mice and obese (ob/ob) mice ($n = 4$), followed by KEGG pathway analysis. The data underlying the graphs shown in the figure can be found in **S1 Data**.
(TIF)

**S7 Fig. Obesity alters hepatic gene expressions associated with Wnt signaling pathway and those of the pericentral or periportal zones.** Lean (purple bars) or ob/ob (yellow bars) mice were used. **(A)** Hepatic gene expressions associated with Wnt signaling pathway, **(B)** those of the pericentral zones, and **(C)** those of the periportal zones ($n = 5$) were examined by quantitative RT-PCR. The data underlying the graphs shown in the figure can be found in **S1 Data**. Data are presented as means ± SD. **$P < 0.01$ by the unpaired $t$ test.
(TIF)

**S8 Fig. Heatmap of metabolome analysis of the liver.** Rspo3 or LacZ adenovirus was administered to standard diet-fed ob/ob mice. Metabolome analysis was performed by using the liver

($n = 4$). The data underlying the heatmap shown in the figure can be found in **S1 Data**.
(TIF)

**S9 Fig. Metabolome analysis of the liver.** Rspo3 (red bars) or LacZ (blue bars) adenovirus was administered to standard diet-fed ob/ob mice. Metabolome analysis was performed by using the liver ($n = 4$) and graphs were obtained from relative areas. The data underlying the graphs shown in the figure can be found in **S1 Data**. Data are presented as means ± SD. $*P < 0.05$, $**P < 0.01$ by the unpaired $t$ test.
(TIF)

**S10 Fig. Gene ontology (GO) analysis of RNA-sequence of the liver.** Rspo3 or LacZ adenovirus was administered to standard diet-fed ob/ob mice. RNA-sequence analysis was performed by using mRNA sample of the liver ($n = 4$), followed by GO analysis. The data underlying the graphs shown in the figure can be found in **S1 Data**.
(TIF)

**S11 Fig. RNA-sequence analysis of the liver.** Rspo3 or LacZ adenovirus was administered to standard diet-fed ob/ob mice. RNA-sequence analysis was performed by using mRNA sample of the liver ($n = 4$). Heatmaps of genes with glucose and lipid metabolic processes were presented. The data underlying the heatmaps shown in the figure can be found in **S1 Data**.
(TIF)

**S12 Fig. Rspo3 locally alters obesity-related features in the liver of obese mice. (A)** Insulin-stimulated serine473 phosphorylation of Akt protein in liver is presented. Immunoblotting with anti-phospho-Akt (Ser473) and Akt antibodies. The relative intensities (the ratios of p-Akt/Akt protein volume) were indicated in the p-Akt panel. **(B)** Hepatic gene expressions associated with glucose and lipid metabolism were examined on day 23 ($n = 4$–5) after adenovirus administration. **(C)** The scheme of a hypothesis was presented that hepatic metabolic homeostasis is maintained with a balance between glucose and lipid. The data underlying the graphs shown in the figure can be found in **S1 Data**. Data are presented as means ± SD. $*P < 0.05$, $**P < 0.01$ by the unpaired $t$ test.
(TIF)

**S13 Fig. Rspo3 improves obesity-related insulin resistance. (A–F)** Rspo3 (red circles) or LacZ (blue circles) adenovirus was administered to diet-induced obese **(A and B)** ($n = 6$), standard diet-fed KK-Ay **(C)** mice ($n = 6$) and standard diet-fed ob/ob **(D and E)** mice ($n = 5$). **(A and C)** Glucose tolerance and **(B)** insulin tolerance tests were performed on day 7 after adenovirus administration. **(D)** Glucose tolerance and **(E)** insulin tolerance tests were performed on day 20 and day 16 after adenovirus administration, respectively ($n = 5$). **(F)** Insulin-stimulated serine473 phosphorylation of Akt protein in skeletal muscle (hindlimb muscles) and WAT were immunoblotted with anti-phospho-Akt (Ser473) antibody, or the individual antibody. The relative intensities (the ratios of p-Akt/Akt protein volume) were indicated in the p-Akt panels. The data underlying the graphs shown in the figure can be found in **S1 Data**. Data are presented as means ± SD. $*P < 0.05$, $**P < 0.01$ by the unpaired $t$ test.
(TIF)

**S14 Fig. Rspo3 decreases c-fos IF intensity in rRPa.** Rspo3 or LacZ adenovirus was administered to standard diet-fed KK-Ay mice. Representative images of HE staining and representative immunofluorescence analysis images of rRPa with anti-c-fos antibody and DAPI staining on day 10 after adenovirus administration. The scale bars indicate 100 μm.
(TIF)

**S15 Fig. Obesity increases adaptive thermogenesis. (A** and **C)** Lean (purple bars) or ob/ob (yellow bars) mice were used. **(B)** Lean (purple bars) or KK-Ay (yellow bars) mice were used. **(A)** Endogenous UCP1 mRNA expressions in BAT ($n = 7$) and epididymal WAT ($n = 5$), and **(C)** VO2 and RQ ($n = 4$–$5$) were examined in lean and ob/ob mice. **(B)** Endogenous UCP1 mRNA expressions in BAT ($n = 7$) and epididymal WAT ($n = 6$–$7$) were examined in lean and KK-Ay mice. The data underlying the graphs shown in the figure can be found in **S1 Data**. Data are presented as means ± SD. *$P < 0.05$, **$P < 0.01$ by the unpaired $t$ test. (TIF)

**S16 Fig. HV does not influence hepatic gene profiles but brocks BAT gene profiles, induced by hepatic Rspo3 induction. (A–G)** Selective hepatic vagotomy (HV) and sham operation (Sham) were performed 7 days prior to the adenovirus administration (Rspo3; red bars and LacZ; blue bars). **(A)** Body weight gain and **(B)** food intake in either SO-LacZ or HV-LacZ mice were examined ($n = 7$–$8$). **(C)** Liver weight ($n = 5$–$8$), **(D)** hepatic LGR5 and Axin2 expressions ($n = 5$–$8$), **(E)** hepatic Hsd17 and Cyp2f2 expressions ($n = 5$–$8$), **(F)** representative immunofluorescence analysis images of the liver in SO- or HV-KK-Ay Rspo3 mice with anti-Rspo3 antibody and DAPI staining, and **(G)** BCAT2 and BCKDH expressions in BAT ($n = 6$–$8$) were examined on day 7 after adenovirus administration. The scale bars indicate 100 μm. The data underlying the graphs shown in the figure can be found in **S1 Data**. Data are presented as means ± SD. *$P < 0.05$, **$P < 0.01$ by the unpaired $t$ test. (TIF)

**S17 Fig. Construction of iLiRspo3KO mice.** The scheme of methods was presented that iLiRspo3KO mice and control mice were created by both CRISPR/CAS9 and Cre-loxP technologies. (TIF)

**S18 Fig. Hepatic gene profiles and metabolic profiles of iLiRspo3KO mice. (A–H** and **J)** iLiRspo3KO mice (pink bars and circles) or control mice (bright green bars and circles) were presented. **(A)** Representative RNA-Scope analysis images of the liver in iLiRspo3KO mice with specific Rspo3 probe. The numbers of Rspo3-positive signals (dots) were measured randomly. **(B)** Food intake ($n = 6$) and body weight gain ($n = 7$–$9$) for 1 week from day 28 or day 4 after tamoxifen administration, respectively. **(C)** Liver weights were measured ($n = 8$–$9$), **(D)** hepatic LGR5, and Axin2, **(E)** Cyp7a1, Rdh9, Oat, and Cyp2f2 mRNA expressions ($n = 7$–$8$), and **(G)** epididymal WAT weight ($n = 8$–$9$) were examined on day 13 after tamoxifen (40 μg/g body weight) administration. **(F)** Glucose tolerance and insulin tolerance tests were performed almost 2 months ($n = 6$–$7$) after tamoxifen (40 μg/g body weight) administration. **(H)** Hepatic Rspo3 expression was examined on day 2 after tamoxifen (120 μg/g body weight) administration ($n = 7$–$8$). **(I)** Representative immunofluorescence analysis images of the pericentral zone in the liver with anti-Rspo3 antibody and DAPI staining. **(J)** Hepatic LGR5, β-catenin, Axin2, Cyp7a1, and Cyp2f2 mRNA expressions were examined on day 2 after tamoxifen (120 μg/g body weight) administration ($n = 6$–$8$). The scale bars indicate 50 μm. The data underlying the graphs shown in the figure can be found in **S1 Data**. Data are presented as means ± SD. *$P < 0.05$, **$P < 0.01$ by the unpaired $t$ test. (TIF)

**S19 Fig. The scheme of a newly identified inter-organ communication system.** (TIF)

**S1 Table. RNA-Sequence analysis of hepatic metabolic genes in the pericentral zones (PC genes) or those in the periportal zones (PP genes).** (TIF)

**S2 Table. RNA-Sequence analysis of hepatic metabolic genes in the pericentral zones (PC genes) or those in the periportal zones (PP genes).**
(TIF)

**S1 Raw Images. Western blot sets used in the article.**
(PDF)

**S1 Data. Raw data supporting Figs 1–6, S3–S13, S15, S16, and S18.**
(XLSX)

**S1 Text. Supporting information about probe, primer, and sequences.**
(DOCX)

# Acknowledgments

The authors thank colleagues and collaborators for their helpful support and advice; J. Hwang and Y. Tezuka for histological analysis; and K. Murata for highly capable research assistance. Finally, the authors gratefully acknowledge the commitment and generosity of the Teikyo University School of Medicine.

# Author Contributions

**Conceptualization:** Kenji Uno.

**Data curation:** Kenji Uno.

**Formal analysis:** Kenji Uno, Takuya Uchino, Takashi Suzuki, Yohei Sayama, Naoki Edo, Kiyoko Uno-Eder, Koji Morita, Toshio Ishikawa, Kazuhisa Tsukamoto.

**Funding acquisition:** Kenji Uno.

**Investigation:** Kenji Uno.

**Methodology:** Kenji Uno.

**Project administration:** Kenji Uno.

**Resources:** Kenji Uno, Miho Koizumi, Hiroaki Honda, Hideki Katagiri.

**Software:** Kenji Uno.

**Supervision:** Kenji Uno.

**Validation:** Kenji Uno.

**Visualization:** Kenji Uno.

**Writing – original draft:** Kenji Uno.

**Writing – review & editing:** Kenji Uno.

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
