## [Editor Report · Decision Letter 0]

30 Aug 2023

Dear Dr Uno, 

Thank you for submitting your manuscript entitled "Rspo3, a mediator of hepatic metabolic zonation, ameliorates diabetes and obesity predisposition via inter-organ communication" for consideration as a Research Article by PLOS Biology.

Your manuscript has now been evaluated by the PLOS Biology editorial staff as well as by an academic editor with relevant expertise and I am writing to let you know that we would like to send your submission out for external peer review.

Once your full submission is complete, your paper will undergo a series of checks in preparation for peer review. After your manuscript has passed the checks it will be sent out for review. To provide the metadata for your submission, please Login to Editorial Manager (https://www.editorialmanager.com/pbiology) within two working days, i.e. by Sep 01 2023 11:59PM.

Kind regards,

Luke

Lucas Smith, Ph.D.

Senior Editor

PLOS Biology

lsmith@plos.org

---

## [Decision Letter · Decision Letter 1]

25 Oct 2023

Dear Dr Uno,

Thank you for your patience while your manuscript "Rspo3, a mediator of hepatic metabolic zonation, ameliorates diabetes and obesity predisposition via inter-organ communication" was peer-reviewed at PLOS Biology, and I apologize again for the delay in sending you a decision. Your study has now been evaluated by the PLOS Biology editors, an Academic Editor with relevant expertise, and by several independent reviewers. 

In light of the reviews, which you will find at the end of this email, we would like to invite you to revise the work to thoroughly address the reviewers' reports. As you will see below, the reviewers find the study interesting, but highlight that more work is needed to fully support the conclusions. 

Given the extent of revision needed, we cannot make a decision about publication until we have seen the revised manuscript and your response to the reviewers' comments. Your revised manuscript is likely to be sent for further evaluation by all or a subset of the reviewers.

We expect to receive your revised manuscript within 3 months, but in this case, we appreciate that that addressing all of the reviewer concerns may require a considerable effort, and we would be happy to extend the deadline for your revision by a few months, if that is helpful. Please email us (plosbiology@plos.org) if you have any questions or concerns, or would like to request an extension.

**IMPORTANT - SUBMITTING YOUR REVISION**

*Re-submission Checklist*

*Published Peer Review*

*PLOS Data Policy*

*Blot and Gel Data Policy*

Sincerely,

Lucas

Lucas Smith, Ph.D.

Senior Editor

PLOS Biology

lsmith@plos.org

REVIEWS:

Reviewer #1: In their manuscript entitled "Rspo3, a mediator of hepatic metabolic zonation, ameliorates diabetes and obesity predisposition via inter-organ communication" Uno and colleagues highlight neuronal and non-neuronal inter-organ linkages to be involved in hepatic Rspo3-mediated maintenance of systemic glucose metabolism and body composition. Therefore, the authors artificially increase Rspo3 expression using an adenovirus-based delivery system in lean and obese mouse models. They show that increased Rspo3 expression ameliorates obesity-associated features, reduces adipose tissue enlargement and restores insulin sensitivity and skeletal muscle quality. By performing a hepatic vagotomy the authors provide some evidence that the effects of Rspo3 include a partial neuronal involvement (ie ameliorated glucose intolerance). Moreover, they identify the hepatic PPARd-to-muscle PPARa-axis to be involved in the regulation of fatty liver development and reduced skeletal muscle quality in obese mice. In line, the authors find these phenotypes to be exacerbated upon reduced hepatic Rspo3 expression.

Overall, Uno et al. highlight an interesting concept linking hepatic WNT signaling which controls liver zonation to the body`s overall health status via inter-organ communication. These findings have the potential to help identifying new therapeutic targets for treating obesity phenotypes. However, several shortcomings of the manuscript should be addressed before publication.

Major comments:

1. The authors claim that Rspo3 expression induced by adenovirus delivery is selective to the liver. However, they never proof this selectivity, and many adenoviral systems are not exclusively targeting the liver. It is a must, to assess Rspo3 expression in other organs (especially adipose tissue and skeletal muscle) to make sure that all observed effects are truly mediated via the liver as claimed by the authors.

2. The authors suggest RSPO3 is mainly expressed in zone3 hepatocytes, whereas many previous studies collectively suggest that WNT ligands, including RSPO3, are mostly expressed in central vein endothelial cells and neighboring sinusoidal endothelial cells. The authors should provide better staining evidence with co-stainings for hepatocyte markers and endothelial cell markers, as well as ISH stainings for RSPO3 with cell type co-stainings to clarify this point. The relatively low reduction in RSPO3 levels in the AlbCreER-induced RSPO3 KO further suggests that the majority of RSPO3 is generated elsewhere.

3. On page 11, lines 7-10 the authors describe upregulation of zone3 and reduction of zone1 metabolic enzymes upon RSPO3 overexpression. This finding is not novel since the role of RSPO3 in metabolic zonation (Rocha et al, Cell Reports 2015) and reprogramming of zone1 into zone3 hepatocytes by RSPO treatment, or deletion of ZNRF3/RNF43, (Sun et al, Cell Stem Cell 2021) have been previously shown. The authors should end that section by referencing these previous findings. Although the authors find exciting new data suggesting RSPO3-related interesting extrahepatic phenotypes, impaired metabolic zonation in patients is major challenge for such a therapy and should be discussed in detail.

4. The authors write in the last sentence on page 11 that RSPO3 slightly increased liver weight, whereas Figure 2E show almost doubled liver weight. RSPO injections increased hepatocyte proliferation and liver weight (Planas-Paz et al, Nature Cell Biology 2016 and Sun et al, Cell Stem Cell 2021) and combined of ZNRF3/RNF43 resulted into uncontrolled hepatocyte proliferation and hepatic tumor formation (Sun et al, Cell Stem Cell 2021). The author have to assess proliferation in their model by Ki67 staining, or if possible by EdU incorporation. Please note the Adenoviral injections induce inflammation which can mask the proliferative effect, so later timepoints (min. 2 weeks post injection) should be assessed. Importantly, the authors have to discuss this caveat associated with potential RSPO3-related therapies. 

5. The authors observed that many metabolic genes are deregulated upon increased Rspo3 expression. Instead of picking single candidate genes, they should use their RNA-seq data to get an overall picture (perform GSEA or GO term analysis) and support their hypotheses on the regulation of entire pathways. Complete lists of PC and PP metabolic genes should be shown as well in line with the previous comment.

6. In Fig3A they claim that Rspo3 expression via adenoviral administration may be lower in obese than in lean mice. This comparison is not shown, thus the authors should either add a corresponding graph or remove this sentence.

7. Fig S4 shows a metabolomics analysis. Although the overview map may be helpful to establish the connections between metabolites for the reader, it is more important to convey the message which pathways are up/downregulated. Therefore, the authors should consider exchanging/adding a heatmap instead and group metabolites by pathway. Also here, metabolic processes related to PP hepatocytes should be shown to check whether RSPO3 expression impaired PP hepatocyte metabolism via reprogramming of PP hepatocytes into PC hepatocytes. 

8. Fig3D should highlight Akt phosphorylation. However, loading controls and quantification are missing (same for Fig3E, as well as for many other WB analyses!). This must be added for all WB. Additionally, Foxo1 translocation can only be claimed when cytoplasmic levels are shown together with nuclear levels. Otherwise, it might be just increased expression. In addition, the authors conclusions must be toned down (decreased gluconeogenesis and glucose utilization in TCA are not shown by the authors).

9. In Fig S5 the authors make assumptions about minor changes of PPARd and CPT1a (lower than 1.5-fold, and not even confirmed on protein levels). If this remains the only evidence, the authors must tone down their conclusions.

10. The sentence after citation 30" thereby leading to the restoration of adrenergic signaling, mitochondrial biogenesis and skeletal muscle mass [31], which had been impaired in the setting of obesity" is an overstatement considering that the authors only refer to increased b2AR expression and resultant PGC1a expression in the muscle. In general, the entire following sections contains a lot of discussion and correlative findings (i.e. referring to PC18:0/18:1). Either this should be moved into the discussion part, or alternatively, the authors should add some more supportive data here.

11. The authors claim that HV also almost completely blocked Rspo3-induced BCAT2 and BCKDH expression. However, for FigS8D significances are not shown when comparing Rspo3 conditions between Sham and HV as done for literally every other gene. Given the error bars for BCAT, this raises doubts regarding the drawn conclusion. Can the authors show the graph displaying single samples at least to the reviewer?

Minor comments:

1. Most histological staining pictures are too small, quality and size should be increased. CVs and PVs should be labelled.

2. The word "ironically" must be removed from the abstract.

3. What do the authors mean by the term "feeding behavior"? The exact timepoint of analysis after feeding should be stated in the text.

4. The authors use the term "Rspo3 supplementation" throughout the text. However, supplementation is rather used in a dietary context and should be exchanged for a more appropriate term (eg use increased Rspo3 expression).

5. The term "as previous report indicating" used several times throughout the text is grammatically incorrect. 

6. Quantification for number of lipid droplets is missing (Fig.3H).

7. FigS5 heatmap should be adjusted to color-blind standards.

8. The graphical abstract in Fig7 is too complicated and should be reduced to the major findings. Alternatively, this extended version could be kept in supplement and a "reduced" version presented in the main figures.

Reviewer #2, Costas Christodoulides (note, Reviewer 2 has signed this review): In this paper, Uno and colleagues investigate the role of hepatic Rspo3 on systemic metabolism using gain- and loss-of-function approaches. Consistent with previous reports, they show that adenovirally mediated, hepatic over-expression of Rspo3 alters liver metabolic zonation (PMID: 26655896) and it is further associated with reduced weight gain and improvements in glucose tolerance and systemic insulin sensitivity in both lean and ob/ob mice. In ob/ob mice, hepatic Rspo3 over-expression also ameliorated hepatic steatosis. In another mouse model of genetic obesity (KK-Ay), hepatic vagotomy appeared to partially reverse the beneficial effects of hepatic Rspo3 over-expression on adiposity and glucose tolerance. Reciprocally, tamoxifen-induced, hepatic knockdown of Rspo3 led to impaired glucose tolerance and systemic insulin sensitivity, as well as, increased hepatic steatosis in chow-fed animals. Based on these findings, the authors postulate that "targeting hepatic metabolic zonation by Rspo3 may provide new therapeutic strategies against diabetes and the metabolic syndrome".

Strengths of the manuscript include that many of the findings are novel and are supported by evidence from both gain- and loss-of-function experiments conducted in vivo and ex vivo. Nonetheless, I have several comments/criticisms as listed below:

Major criticisms:

1. The authors examine only the short-term effects (one week) of hepatic Rspo3 gain- and loss-of-function on adiposity and systemic metabolism (see also comment 6).

2. Based on gene expression analyses, the hepatic over-expression of Rspo3 achieved with adenoviral transductions was very high (10-20x) and was associated with a significant increase in liver size e.g., almost doubling in chow-fed animals (fig. 2e).

3. The authors show in figure 1, that Rspo3 circulates in plasma and furthermore, that it's circulating levels are up-regulated in the fed state and dysregulated in obesity. Yet, they fail to show any data on circulating Rspo3 levels in their gain- and loss-of-function animal models. Additionally, they fail to consider that any of the effects of hepatic Rspo3 on weight gain and systemic metabolism could be mediated by endocrine actions of Rspo3, which I find quite puzzling. 

4. Another major limitation, is that the authors failed to measure food intake and energy expenditure in any of their mouse models. This makes it difficult to interpret and draw firm conclusions from their findings.

5. The abstract and results sections should be factual. Yet, they contain a lot of speculation. For example, the authors claim that a remote, lipid mediated, liver-muscle communication system mediates some of the beneficial effects of hepatic Rspo3 on systemic metabolism and improves skeletal muscle quality. However, the data presented fail to directly demonstrate such an axis. Specifically, the authors do not demonstrate either increased circulating phosphatidylcholine levels in their Rspo3 gain-of-function animal models, or direct effects of phosphatidylcholine on skeletal muscle gene expression, or that their animals have e.g., increased exercise tolerance. Thus these comments are best included as speculation in the discussion. There are multiple other similar examples in the results section. 

6. The authors do not provide any direct evidence that the activity of the sympathetic nervous system in adipose tissue is altered in response to changes in hepatic Rspo3 expression, yet this is one of their main conclusions. 

7. Rspo over-activation has been linked with cancer e.g., RSPO3 gene fusions which are associated with increased RSPO3 expression have been identified in colon cancer patients (PMID: 34663878). Consistently, transgenic mice with increased circulating Rspo levels displayed massive intestinal enlargement in mice (PMID: 16109882, 16357527). On the other hand, Rspo are important for stem cell maintenance and proliferation (PMID: 22617424). Hence, it is highly likely that manipulation of hepatic Rspo3 expression for metabolic disease prevention/treatment purposes will have a very narrow therapeutic window (if at all possible).

Specific points:

1. Could energy repartitioning to the growing liver, and potentially growing intestines, contribute to the beneficial effects of hepatic Rspo3 over-expression on systemic metabolism?

2. The authors claim that "Rspo3 supplementation restores the hyperactivated metabolic rates of BAT-derived adaptive thermogenesis against the development of obesity". However, they do not provide any evidence that BAT is overactive in their ob/ob mice. Secondly, they do not examine any direct effects of Rspo3 on brown adipocyte gene expression. Finally, it is difficult to believe that reduced BAT activity contributes to the reduced weight gain and improved systemic metabolism of mice with hepatic over-expression of Rspo3.

3. I would like to see the effects of hepatic vagotomy on animal weights in figure 5. It is also curious that figure 5D shows data on the ratio of epiWAT/body weight in the vagotomised animals whereas figures 2I and 4D compare the effects of hepatic Rspo3 over-expression on epi WAT weight uncorrected for total body weight.

4. Did hepatic vagotomy have any effects on WAT Ucp1 and Pgc1a gene expression or on systematic insulin sensitivity as assessed by insulin tolerance tests?

5. Did Rspo3 knockdown have any effects on liver size, weight gain, or epididymal WAT weight?

6. Ideally, the authors should quantify the hepatitic lipid content in animals with Rspo3 knockdown and over-expression.

Other points:

1. In figure 1C, was Rspo3 expression reduced in BAT and skeletal muscle from obese mice?

2. Is beta-catenin a WNT target gene?

3. With reference to the data in figure 3E, does PDK4 primarily act in the nucleus?

Reviewer #3: This manuscript by Uno et al reports the role of Rspo3 that is known to determine hepatic zonation in regulating glucose and energy homeostasis. They show that Rspo3 increases with feeding and this rise is attenuated with insulin resistance such as in ob/ob mice. Histologically, they show that Rspo3 is further concentrated in the periportal area and this does not occur in models of insulin resistance. Further, in metabolic tissues such as the liver or adipose tissue. Rspo gene expression progressively declines with insulin resistance such as with high fat diet feeing or in ob/ob mice. Next, they deliver Rspo3 via adenovirus and show increased expression in the liver in addition to improved metabolic parameters such as fasting insulin and glucose levels as well as gluconeogenic genes. This is done also in ob/ob mice showing similar metabolic benefit 7 days post gene delivery. They then examine the mice with deletion of Rspo3 in liver and show metabolic exacerbation. Similar to the overexpression data vi adenovirus, deletion model is also examined after a short term of 7 and 13 days post deletion. Again, this is a short duration given the chronic nature of the diseases being examined and longer term data would be required. Overall, the data presented are interesting and they use multiple in vivo models as well as overexpression and deletion models to show the importance of Rspo3 in metabolic regulation. However, throughout the manuscript, the data interpretation is far-reaching and many of their statements are not at all supported by the data presented as follows. It is suggested that the authors more accurately describe the data without overreaching with conjecture. In particular, much of what is shown as a working model in Fig 7 is not at all shown in their data and would strongly suggest revising to only show what is well-supported by their data.

What is the difference between data presented in Fig 1B versus 1C, and why are the data different for the liver shown in B versus C? 

Fig 1E needs to be shown in a lower magnification that demonstrate the zonation more clearly. The image shown in the figure does not capture the peri-central zonation convincingly. 

Fig 2A shows increased Rspo3 levels in gene and protein expression. The increase in protein is unclear from the western blot shown and should be quantified. The degree of increase in gene versus protein expression is not at all congruent and expression in other tissues need to be shown. Furthermore, histologic imaging needs to be shown to test whether the increased expression of Rspo3 is more enriched pericentrally as shown with endogenous Rspo3. 

What is the evidence that 'hepatic Rspo3 induction exerted remote effects, i.e., the weight of epididymal WAT was significantly decreased (Fig 2I),' - they don't have data to support this statement. The effect in the adipose tissue may be from its expression in this tissue and not the distant effect of the liver, especially since they've already shown expression of Rspo3 in adipose tissue (Fig 1C). Also, in this same section, their postulation involving the sympathetic activation is completely not supported by their data and should be removed.

All the metabolic experiments are done 7 days post gene delivery. Given the chronic nature of the metabolic conditions such as diabetes and obesity that they are investigating, more longer term experiments need to be done as well as gene dosage effect. How did they come up with the dose of gene to be delivered. What are the expression levels of the gene delivery in other tissues such as the adipose tissue, muscle, brain or the immune cells? These are all tissues that can affect the metabolic outcome.

Similar comments re duration after knockout apply as stated above. 7 and 13 days post knockout is not sufficient to gain insight into the chronic nature of the diseases being examined. 

For knockdown experiments, why only 20%? This needs to be explained. Is this because more knockdown would lead to severe hepatic derangements? This is an important issue that needs to be addressed and shown with more significant knockdown.

There are multiple inappropriate descriptions of their data or of the subject matter that they describe as follows:

Abstract - pg 4 - Rspo3 is not a 'bioactive substance' - they have manipulated gene expression using adenovirus to investigate its role - this needs to be accurately described in the abstract and throughout the manuscript.

When describing data in Fig 1D, they should be insulin 'responsive' tissue - , not insulin 'reactive'. Also, it's not feeding 'behaviour'. It's simply 'feeding' that would lead to metabolic abnormalities.

'Hepatic Rspo3 supplementation' would not be the best way to describe their experiment. Supplementation has a nutrition connotation and should simply describe their systemic gene delivery rather than 'supplementation'. This is used throughout the section describing findings for Fig 3, which is advised to be removed.

'hepatic PPARd-to-muscle PPARa axis' - again is overstretch of data interpretation.

---

## [Decision Letter · Decision Letter 2]

10 Jun 2024

Dear Dr Uno,

Thank you for your patience while we considered your revised manuscript "Rspo3, a mediator of hepatic metabolic zonation, ameliorates diabetes and obesity predisposition via inter-organ communication" for publication as a Research Article at PLOS Biology. Your revised study has been evaluated by the PLOS Biology editors, the Academic Editor and the original reviewers.

As you will see below, the reviewers agree that the study has been strengthened in this revision and reviewer 1 has suggested we accept the study. However, both reviewers 2 and 3 have a number of important lingering concerns that we think should be addressed, including with new data, in another revision before we can consider your study for publication. 

We do note that the tone of reviewer 3's comments is quite critical, but after discussing his/her comments with the other reviewers and the Academic Editor, we agree with the substance of many of the concerns raised by this reviewer, including that a number of conclusions have been over-interpreted and are not fully supported. In many places we think that it may be possible to address reviewer 3's comments via textual changes - by toning down some conclusions, reworking the abstract, and adding additional explicit discussions of limitations. We would not require the generation of additional long-term metabolic phenotyping of mice with hepatic RSPO3 expression, as we think that this is beyond the scope of the current work, and there is still value in the current findings without that data. But we think the absence of this data should be discussed as a limitation. 

With that said, we also think that new data and analyses will be required to address a number of the concerns raised by Reviewer 2 and 3. While we tend not to invite a second experimental revision, given our and the reviewer interest in the study, we are willing to do so here.

Given the extent of revision needed, we cannot make a decision about publication until we have seen the revised manuscript and your response to the reviewers' comments. Your revised manuscript is likely to be sent for further evaluation by all or a subset of the reviewers.

**IMPORTANT - SUBMITTING YOUR REVISION**

*Re-submission Checklist*

*Published Peer Review*

*PLOS Data Policy*

*Blot and Gel Data Policy*

Sincerely,

Luke

Lucas Smith, Ph.D.

Senior Editor

PLOS Biology

lsmith@plos.org

REVIEWS:

Reviewer #1, Jan Tchorz (note Reviewer 1 has signed this review): The authors addressed all my comments and should be congratulated to an excellent paper that will certainly get significant attention. Jan Tchorz

Reviewer #2, Costas Christodoulides (note, Reviewer 2 has signed this review): The authors have adequately addressed most of my concerns. However, I do have a few additional comments.

Major Comments:

 1. The authors state that modulating hepatic RSPO3 expression may "restore metabolic zonation." Is there any evidence, such as transcriptomic data either generated by the authors or publicly available, that hepatic zonation is disrupted in mouse models of dietary and genetically induced obesity?

 2. Effects of Vagotomy: The authors claim that a neural inter-organ communication system is involved in hepatic RSPO3-induced actions in adipose tissues and skeletal muscle. However, according to the data in Supplementary Figure 12A, vagotomy abrogates the weight loss associated with hepatic RSPO3 overexpression in genetically obese mice. Could this be the primary reason for the gene expression changes in adipose tissues and skeletal muscle, as well as the changes in glucose tolerance and insulin sensitivity in hepatic RSPO3-overexpressing mice post-vagotomy? Also, I would like to see the effects of hepatic vagotomy on fasting glucose levels. Is it possible that hepatic vagotomy increases hepatic glucose output, thereby potentially contributing to a switch from fat to glucose metabolism in peripheral tissues in vagotomised RSPO3-expressing mice?

 3. In Figure 5, panels D, E, G, and J, please also include data from control LacZ mice.

Other Comments:

 1. Hepatic RSPO3 expression in lean animals approximately leads to a doubling of liver weight. This is not a slight change as stated by the authors (page 12, second from bottom line).

 2. Duration of Metabolic Benefits: Additionally, 20 days is not a "relatively long period" and should not be reported as such by the authors. They should simply state that metabolic benefits lasted up to 20 days.

 3. Page 13, bottom of the second paragraph. The authors state that increased beta3AR, UCP1, and PGC1a expression in epididymal WAT contributes to decreased body weight after 7 days of adenoviral treatment. This is speculation. We are not shown any calorimetry data to support this. In fact, based on the authors' data, the weight loss in these animals is driven by reduced appetite.

 4. Page 16, second paragraph. The authors mention that hepatic RSPO3 induction regulates hepatic lipogenesis, but the only support for this is changes in gene expression of lipogenic genes. Hence, this statement should be tempered.

 5. Page 20, top paragraph: "This so-called energy-saving machinery in obese …. with regulating BAT thermogenic physiology." This is speculation, hence this statement should be tempered and ideally moved to the discussion.

 6. Page 20, bottom paragraph: "Hepatic RSPO3 induction…. is impaired in the setting of obesity." Again, this is speculation as the authors have not shown that adrenergic signaling, mitochondrial biogenesis, skeletal muscle mass, and muscle remodeling are improved consequent to hepatic RSPO3 expression. They just show gene expression changes.

 7. In Figure 5I, the effects of vagotomy on norepinephrine content in WAT seem to be driven by an increased content in the LacZ mice rather than a decrease in the mice with hepatic RSPO3 overexpression.

 8. The immunofluorescence images are of poor quality, and I suggest using higher resolution figures in any publication.

 9. The authors claim that Figure 1C shows that RSPO3 expression is reduced in BAT and skeletal muscle of obese mice. However, the bar charts seem to suggest the opposite.

 10. Beta-catenin is post-transcriptionally, rather than transcriptionally, regulated by WNT/beta-catenin signaling. Unlike Axin2 and LGR5, it is not a beta-catenin target gene, and the references quoted by the authors do not support such a premise.

Reviewer #3: This revised manuscript shows some improvement and the authors have attended to some concerns raised by the reviewers. They improved on systematically showing expression patterns of Rspo3 in the liver under various metabolic conditions and in response to feeling and fasting. They have used several models of obesity to show the importance of RSPO3 with overexpression and knockout. However, the study continues to have major concerns around their interpretation of data and overreaching of conclusions that are often stated even before data description in the results section. Major issue is around the acute short-term analysis for a disease model that is long term that limits the significance of the study. They have expanded only some models to 21 days which is still insufficient and also, would be important to show in a longitudinal fashion so that we can get some sense of continual/plateau versus temporary effect. 

The following are some specific comments:

Abstract - 

The metabolic zonation in the liver is thereby broadly defined as 'spatiotemporal'

variability… it is not spaciotemporal - rather it's a 'spatial' phenomenon - but not temporal - thus the term zonation. This elaboration is not needed in the abstract in particular.

'Herein, we demonstrate that altered hepatic metabolic zonation due to Rspo3 contributes to

maintaining systemic glucose metabolism and body composition via a new mechanism

involving neuronal and non-neuronal inter-organ linkages' - this also is not a statement that should appear in the abstract before data description - pls remove and consider adding at the end if supported by overall data.

Similarly - the following sentence are not data description but rather conclusions. 'Hepatic Rspo3 induction in obesity improves obesity-associated features, e.g., fatty liver, insulin resistance and diabetes, by reversing adipose tissue enlargement and restoring attenuated organ insulin sensitivities, overstimulated adaptive thermogenesis and decreased skeletal muscle quality.' - It would be best to describe data. 

The entire abstract is unacceptably composed of interpretation/conjecture rather than data description - would be best to describe/summarize data followed by overall conclusion.

Pg 7 - high-fat diet(s) - should be high fat diet 

Fig 1 - Nice work showing systematic data beginning with expression pattern in models of obesity and in lean conditions both fasted and fed. However, to conclude this section with "Therefore, targeting hepatic Rspo3 may allow new mechanisms involved in regulating

metabolic homeostasis in the liver to be identified, which could in turn lead to the

detection of remote effects on systemic metabolism with physiological (against obesity)

or pathophysiological (promoting obesity) significance. Pg 11' is not at all appropriate and would be better to summarize their data.

'we expressed Rspo3' on pg 11 should be 'overexpressed'

For fig 2A 2nd panel Y-axis label - should be 'relative intensity'?

For fig 2 - they show effects after 7 days and day 20 is only shown in S3 without corresponding weight data and also no description of this data in the results section. Body weight data needs to be shown in a longitudinal fashion with absolute weights rather than relative change. Metabolic expression data shown in fig 2 after 7 days also needs to be shown after 20 days - (genes shown in fig 2D).

Fig 2A does not show 'selective transgene expression in the liver' pg 11. For this statement to be supported, they need to show expression in other tissues.

'These results indicated hepatic Rspo3 induction to up-regulate the Rspo3/Wnt signaling pathway in the liver and thereby shifted metabolic gene expressions, a process by which hepatic features of metabolism are considered to move from the periportal to the pericentral zone. These phenomena, as described above, are compatible with those outlined in previous reports which indicated the roles of Rspo3 in regulating metabolic zonation [13] or reprograming of zonation in the liver [18]. (pg 12) is not supported by their data preceding this conclusion.

To examine proliferative capacity of the liver in response to Rspo3 overexpression, Ki67 should be shown histologically. It is not clear what the proliferative impact of Rspo3 overexpression is, particularly given the known impact in malignancy. This might explain the increase in liver weight.

Fig 3 - using ob/ob - again - only examined after 7 days with minimal difference in body weight - absolute body weight in a longitudinal fashion need to be shown for the two groups rather than a relative change as shown in Fig 3M. 

'Rspo3 induction increased Ki67 IF activity (S4H and S4I Fig)' (pg 15) is not true-. IF does not measure activity.

p-Akt in the livers of ob/ob mice (Fig 3D) needs to be quantified and need more n. n=2 with intensity shown above that ranges widely and is not sufficient.

These improvements of glucose 'metabolism' (pg 18) - should be 'tolerance'

murine models (S11A and S11B Fig), resulting increased resting VO2 as compared to that

of lean mice (S11C Fig) - should also include once overexpressing Rspo3. Otherwise these data are not helpful for their manuscript.

'This so-called energy-saving machinery in obese Rspo3-mice is partly explained by hepatic GK up-regulation, in which hepatic GK mediates an inter-organ linkage associated with regulating BAT thermogenic physiology [30]. These results raise the possibility that the reconstitution of hepatic metabolic zonation by Rspo3 induction in obesity, with up-regulation of WAT-derived

thermogenesis and restoration of overstimulated BAT-derived thermogenesis, leads to

balancing and redistribution of thermogenesis derived from different adipose tissues such

as WAT and BAT.' (pg 20) - would advise removing from results section. These are all expected findings with protection from weight gain. To implicate adrenergic nerve involvement etc is a complete overstretch. Would advise involving a scientist in metabolism to properly interpret these results. 

Implication of vagal nerve in RSPO3 action is not at all clear nor with good rationale. Furthermore, the data presented are with large error bars and not meaningful. HV is not at al helpful in the results and adds to the confusion and distracts the main story. Would suggest removing. It reflects the naïve understanding of their data.

The scheme shown in S15 is not at all supported by their data and reflects very poorly on their understanding of their data. Would advise removing any neuronal involvement and keeping purely the data in the different tissues that are known effects of improvement in hepatic insulin sensitivity.

For the KO data, decrease of less than 20% expression with such phenotype is highly questionable and hard to explain. Also, their IF data (S14I) does not show deletion to any appreciable degree. Also, short term data bring limited clinical relevance with no body weight data.

---

## [Decision Letter · Decision Letter 3]

6 Nov 2024

Dear Dr Uno,

Thank you for your patience while we considered your revised manuscript "Rspo3, a mediator of hepatic metabolic zonation, ameliorates diabetes and obesity predisposition via inter-organ communication" for publication as a Research Article at PLOS Biology. This revised version of your manuscript has been evaluated by the PLOS Biology editors, the Academic Editor and one of the original reviewers. I apologize for the unusual delay incurred while we attempted to obtain additional advice from the reviewers.

Based on the reviews and on our Academic Editor's assessment of your revision, we are likely to accept this manuscript for publication, provided you satisfactorily address the following data and other policy-related requests.

IMPORTANT:

1) Please change your title to: "Rspo3-mediated metabolic liver zonation regulates systemic glucose metabolism and body mass in mice"

2) Please provide grant numbers for all the grants mentioned.

3) Please include the full name of the IACUC/ethics committee that reviewed and approved the animal care and use protocol/permit/project license. Please also include an approval number.

4) Please modify your abstract to include that this work has been performed in mice. We would also like to suggest you to further edit the abstract, to include more details on the experiments that are performed. We suggest "in this study, we analyse the function of Rspo3 in mice"

-change 'expression patterns of endogenous Rspo3' to 'Rspo3 expression analysis showed that Rspo3 expression patterns are spatiotemporally controlled'

-change 'Hepatic Rspo3 induction in obesity' to 'We find that viral-mediated induction of Rspo3 in hepatic tissue' 

Please ensure that you adhere to our data policy http://journals.plos.org/plosbiology/s/data-availability

5) please ensure that the transcriptomics and metabolomic data are deposited in public repositories and provide us with the DOI.

6) Please ensure that you provide the individual numerical values that underlie the summary data displayed in the following figure panels as they are essential for readers to assess your analysis and to reproduce it:

Fig. 1A-D, G, I; Fig. 2A-E, G-M; Fig. 3A-G, I-N; Fig. 4A, B, D-I, K, L; Fig. 5A-E, G, I-L; Fig. 6A-D, F-K, M; Fig. S3A-H; Fig. S4A-F, I; Fig. S5; Fig. S6; Fig. S7A-C; Fig. S8; Fig. S10; Fig. S11; Fig. S12B; Fig. S13A-E; Fig. S15A-C; Fig. S16A-E, G and Fig. S18A-G, H, J

We expect to receive your revised manuscript within two weeks. 

*Published Peer Review History*

*Press*

Sincerely,

Suzanne

Suzanne De Bruijn, PhD 

Associate Editor

sbruijn@plos.org

PLOS Biology

DATA POLICY:

CODE POLICY

Per journal policy, if you have generated any custom code during the course of this investigation, please make it available without restrictions. Please ensure that the code is sufficiently well documented and reusable, and that your Data Statement in the Editorial Manager submission system accurately describes where your code can be found. [IF APPLICABLE: As the code that you have generated to XXX is important to support the conclusions of your manuscript, its deposition is required for acceptance.]

Reviewer remarks:

Reviewer #2: The authors have adequately addressed my comments.

---

## [Editor Report · Decision Letter 4]

27 Nov 2024

Dear Kenji,

Thank you for the submission of your revised Research Article "Rspo3-mediated metabolic liver zonation regulates systemic glucose metabolism and body mass in mice" for publication in PLOS Biology, and thank you for addressing our last editorial requests in this revision. On behalf of my colleagues and the Academic Editor, Jason W. Locasale, I am pleased to say that we can in principle accept your manuscript for publication, provided you address any remaining formatting and reporting issues. These will be detailed in an email you should receive within 2-3 business days from our colleagues in the journal operations team; no action is required from you until then. Please note that we will not be able to formally accept your manuscript and schedule it for publication until you have completed any requested changes.

**IMPORTANT: Please note, that before accepting your study I have updated the data availability statement (DAS) and the supplementary western blot data file, after discussing these with you over email. 

The DAS now reads: 

"All relevant data are within the paper and its Supporting Information files. The original data for the RNA-sequencing experiments have been deposited on the Gene Expression Omnibus (GEO accession GSE282890). Metabolomics data has been deposited to the Metabolomics Workbench (accession number ST003586).

Further requests for resources and reagents should be directed to and will be fulfilled by the corresponding author, Kenji Uno."

>>Please do carefully look through the manuscript file, and double check that the new DAS and western blot file looks good to you after these changes. 

PRESS

Sincerely, 

Lucas Smith, Ph.D.

Senior Editor

PLOS Biology

lsmith@plos.org